# Delayed induction of type I and III interferons mediates nasal epithelial cell permissiveness to SARS-CoV-2

Catherine F. Hatton [1,13], Rachel A. Botting [2,13], Maria Emilia Dueñas [2,13], Iram J. Haq [1,3,13], Bernard Verdon [2,13], Benjamin J. Thompson [1], Jarmila Stremenova Spegarova [1], Florian Gothe [1,4], Emily Stephenson[2], Aaron I. Gardner [1], Sandra Murphy [2], Jonathan Scott[1], James P. Garnett [1], Sean Carrie[5], Jason Powell [1], C. M. Anjam Khan [2], Lei Huang [1], Rafiqul Hussain[6], Jonathan Coxhead [6], Tracey Davey[7], A. John Simpson [1], Muzlifah Haniffa [2,8,9,10], Sophie Hambleton [1,11], Malcolm Brodlie [1,3,14], Chris Ward [1,14], Matthias Trost[2,14], Gary Reynolds[2,14] & Christopher J. A. Duncan [1,12,14✉]

The nasal epithelium is a plausible entry point for SARS-CoV-2, a site of pathogenesis and transmission, and may initiate the host response to SARS-CoV-2. Antiviral interferon (IFN) responses are critical to outcome of SARS-CoV-2. Yet little is known about the interaction between SARS-CoV-2 and innate immunity in this tissue. Here we apply single-cell RNA sequencing and proteomics to a primary cell model of human nasal epithelium differentiated at air-liquid interface. SARS-CoV-2 demonstrates widespread tropism for nasal epithelial cell types. The host response is dominated by type I and III IFNs and interferon-stimulated gene products. This response is notably delayed in onset relative to viral gene expression and compared to other respiratory viruses. Nevertheless, once established, the paracrine IFN response begins to impact on SARS-CoV-2 replication. When provided prior to infection, recombinant IFNβ or IFNλ1 induces an efficient antiviral state that potently restricts SARS-CoV-2 viral replication, preserving epithelial barrier integrity. These data imply that the IFN-I/III response to SARS-CoV-2 initiates in the nasal airway and suggest nasal delivery of recombinant IFNs to be a potential chemoprophylactic strategy.

[1] Translational and Clinical Research Institute, Faculty of Medical Sciences, Newcastle University, Newcastle upon Tyne, UK. [2] Biosciences Institute, Newcastle University, Newcastle upon Tyne, UK. [3] Paediatric Respiratory Medicine, Great North Children's Hospital, Newcastle upon Tyne Hospitals NHS Foundation Trust, Newcastle upon Tyne, UK. [4] Department of Pediatrics, Dr. von Hauner Children's Hospital, University Hospital, Ludwig-Maximilians-Universität Munich, Munich, Germany. [5] Population Health Sciences Institute, Newcastle University, Newcastle upon Tyne, UK. [6] Genomics Core Facility, Biosciences Institute, Newcastle University, Newcastle upon Tyne, UK. [7] Electron Microscopy Research Services, Newcastle University, Newcastle upon Tyne, UK. [8] NIHR Newcastle Biomedical Research Centre, Newcastle Hospitals NHS Foundation Trust, Newcastle upon Tyne, UK. [9] Department of Dermatology, Newcastle Hospitals NHS Foundation Trust, Newcastle upon Tyne, UK. [10] Wellcome Sanger Institute, Wellcome Genome Campus, Cambridge, UK. [11] Great North Children's Hospital, Newcastle upon Tyne Hospitals NHS Foundation Trust, Newcastle upon Tyne, UK. [12] Department of Infection and Tropical Medicine, Newcastle upon Tyne Hospitals NHS Foundation Trust, Newcastle upon Tyne, UK. [13] These authors contributed equally: Catherine F. Hatton, Rachel A. Botting, Maria Emilia Dueñas, Iram J. Haq, Bernard Verdon. [14] These authors jointly supervised this work: Malcolm Brodlie, Chris Ward, Matthias Trost, Gary Reynolds, Christopher J. A. Duncan. ✉email: christopher.duncan@ncl.ac.uk

SARS-CoV-2 is an emergent betacoronavirus responsible for coronavirus disease-19 (COVID-19)[1]. Since its identification in late 2019, global pandemic transmission of SARS-CoV-2 has resulted in over 258 million confirmed infections and ~5.2 million deaths. SARS-CoV-2 infects target cells via the entry receptor ACE2[2] leading to a spectrum of clinical outcomes, ranging from asymptomatic infection to death[3]. Although multiple host factors (e.g. age, male sex, obesity) contribute to adverse clinical outcome[4], the immune response is also decisive, evidenced by the therapeutic benefit of immunomodulatory agents including corticosteroids[5] or IL6 inhibition[6]. Yet much remains to be understood about the immunopathogenesis of COVID-19. Identification of the cells hosting viral entry and characterisation of their response to infection is essential to understanding pathogenesis and improving therapy.

The nasal epithelium is believed to be a key entry point of SARS-CoV-2. Nasal epithelial tropism and efficient viral shedding from the nasopharynx apparently contributes to the high transmissibility of SARS-CoV-2[7], as well as to pathologic features such as anosmia[8]. As an early viral target cell, nasal epithelial cells may also set the tone for the systemic immune response, potentially influencing disease outcome[9]. These factors emphasise the need to study host-virus interaction in human nasal cells. Ex vivo single-cell transcriptomic studies indicate that ciliated and/or goblet cells in the nasal mucosa express ACE2 and TMPRSS2, implicating them as probable SARS-CoV-2 target cells[10,11]. This has been confirmed by in vitro and in vivo studies demonstrating SARS-CoV-2 infection of human nasal epithelial cells[12–15]. Single-cell studies also showed that nasal cells exhibit basal expression of an antiviral expression programme, characterised by induction of several interferon-stimulated genes (ISGs), suggesting that they may be primed to respond to viral infection[10,11]. Interestingly, ACE2 is also regulated by interferons (IFNs)[11,16], implying a complex relationship between IFN signalling and tropism. Type I and type III IFN (IFN-I/III) systems are critical to human antiviral innate immunity[17] and have been implicated in defence against SARS-CoV-2 susceptibility to severe or life-threatening COVID-19 is associated with deleterious variants in IFNAR genes[18,19] and IFN-I blocking autoantibodies[20]. In vitro, SARS-CoV-2 appears sensitive to the antiviral properties of IFN-I, at least in cell lines[21,22], and this activity extends to in vivo model systems[9]. These findings motivate studies to improve understanding of the interaction between SARS-CoV-2 and the IFN-I system in primary human target cells, providing impetus to clinical trials of recombinant IFNs in treatment or prophylaxis of COVID-19[23].

Organotypic cultures of primary human nasal epithelium differentiated at air–liquid interface (ALI) are a translationally relevant primary cell model for studies of SARS-CoV-2 host-virus interaction[12], with considerable potential to accelerate our understanding of pathogenesis. A small number of studies using this model demonstrate that SARS-CoV-2 replicates efficiently in human nasal cells[12–14], yet important questions concerning cellular tropism and their innate immune response remain unresolved. Hou and colleagues report that only ciliated cells were permissive to SARS-CoV-2, despite expression of ACE2 and TMPRSS2 by all cell types[13]. They hypothesised that post-entry factors, such as innate immunity, might govern tropism. By contrast, Pizzorno and colleagues reported infection in all major cell types (ciliated, secretory and basal cells)[14], consistent with prior indications from single-cell RNA sequencing (scRNA-seq) data and studies in lower airway models[24,25]. While an IFN response to SARS-CoV-2 can be detected in nasal cells[12,14], in apparent contrast to bronchial or alveolar epithelial cells[26–28], the kinetics of induction and the antiviral function of IFNs in nasal epithelium has not been systematically characterised.

Here we employ a comprehensive range of techniques, including scRNA-seq and proteomics, in primary human nasal ALI cultures to define: (i) cellular tropism; (ii) the innate immune response to SARS-CoV-2; and (iii) the antiviral activity of IFN-I/III. We observe broad cellular tropism of SARS-CoV-2 for nasal epithelial cell types, although secretory and ciliated cells are the most permissive. Nasal cells mount a delayed IFN response that begins to exert control over viral replication at later times post-infection. However, SARS-CoV-2 remains highly sensitive to IFN-restriction if exogenous IFN-I/III is applied prior to infection. These data improve our understanding of the interaction of SARS-CoV-2 and the human IFN system at the earliest point of infection, with immediate therapeutic implications.

## Results

**SARS-CoV-2 robustly infects primary differentiated nasal epithelial cultures.** Primary nasal epithelial cultures were established from cryopreserved stocks from six adult donors, obtained prior to the SARS-CoV-2 pandemic. Cells were expanded, differentiated and then matured at ALI for 28 days, according to an established protocol[29]. We first sought to address their suitability as a model of SARS-CoV-2 infection. Single-cell RNA-seq libraries were generated from two representative donors, yielding 28,346 individual transcriptomes for analysis following quality control (Supplementary Fig. 1, Supplementary Table 1). Following dimensionality reduction and Leiden clustering, eight populations were discerned by their expression of established marker genes[11,30] (Fig. 1a, Supplementary Data 1). This annotation was further validated using Seurat label transfer from a published scRNA-seq dataset from nasopharyngeal swabs[15] (Fig. 1b). The major populations identified were ciliated, secretory, goblet and basal cells, alongside two rarer populations of FOXN4+ deuterosomal cells[31] and ionocytes. Cells expressed characteristic markers (Fig. 1c and Supplementary Data 1), corresponding closely to ex vivo single-cell data from nasal brushings[15] (Fig. 1b). Immunostaining verified the presence of major cell types in these cultures using well-established protein markers[13]—including acetylated alpha-tubulin-positive (AAT) ciliated cells, mucin 5B (MUC5B)-positive secretory cells, mucin 5AC (MUC5AC)-positive goblet cells, and tumour protein 63 (TP63)-positive basal cells (Supplementary Fig. 2)—with ciliated cells the most abundant population. Consistent with published scRNA-seq data[10,11,15], mRNA for key SARS-CoV-2 entry receptors, ACE2 and TMPRSS2, was expressed, albeit at relatively low levels, alongside other genes implicated in SARS-CoV-2 entry such as FURIN and CTSL (Supplementary Fig. 3)[32]. Robust expression of ACE2 and TMPRSS2 at the protein level was confirmed by immunoblotting of whole-cell lysates prepared from mature ALI cultures (Fig. 1d). To establish their permissiveness to infection, nasal ALI cultures were inoculated at the apical surface with a clinical SARS-CoV-2 isolate (BetaCoV/England/2/20) at an approximate multiplicity of infection (MOI) 0.1—consistent with other studies (0.1–0.5)[12–14]—and monitored for infection over the next 72 h. Expression of SARS-CoV-2 *nucleocapsid (N)* gene and spike (S) protein increased significantly over time, indicative of viral replication (Fig. 1e, f). This was accompanied by the release of infectious viral particles, as measured by plaque assay of apical washes on Vero E6 cells, confirming productive infection (Fig. 1g). SARS-CoV-2 replication was accompanied by a progressive decline in epithelial barrier integrity starting from 48 h post-infection (hpi), reflecting virus-induced epithelial dysfunction and/or potential cytopathic effect (Fig. 1h). These data established the suitability of the human nasal ALI system for modelling SARS-CoV-2 infection.

**Evidence of broad cellular tropism of SARS-CoV-2.** To revisit the question of whether individual cell types are more or less

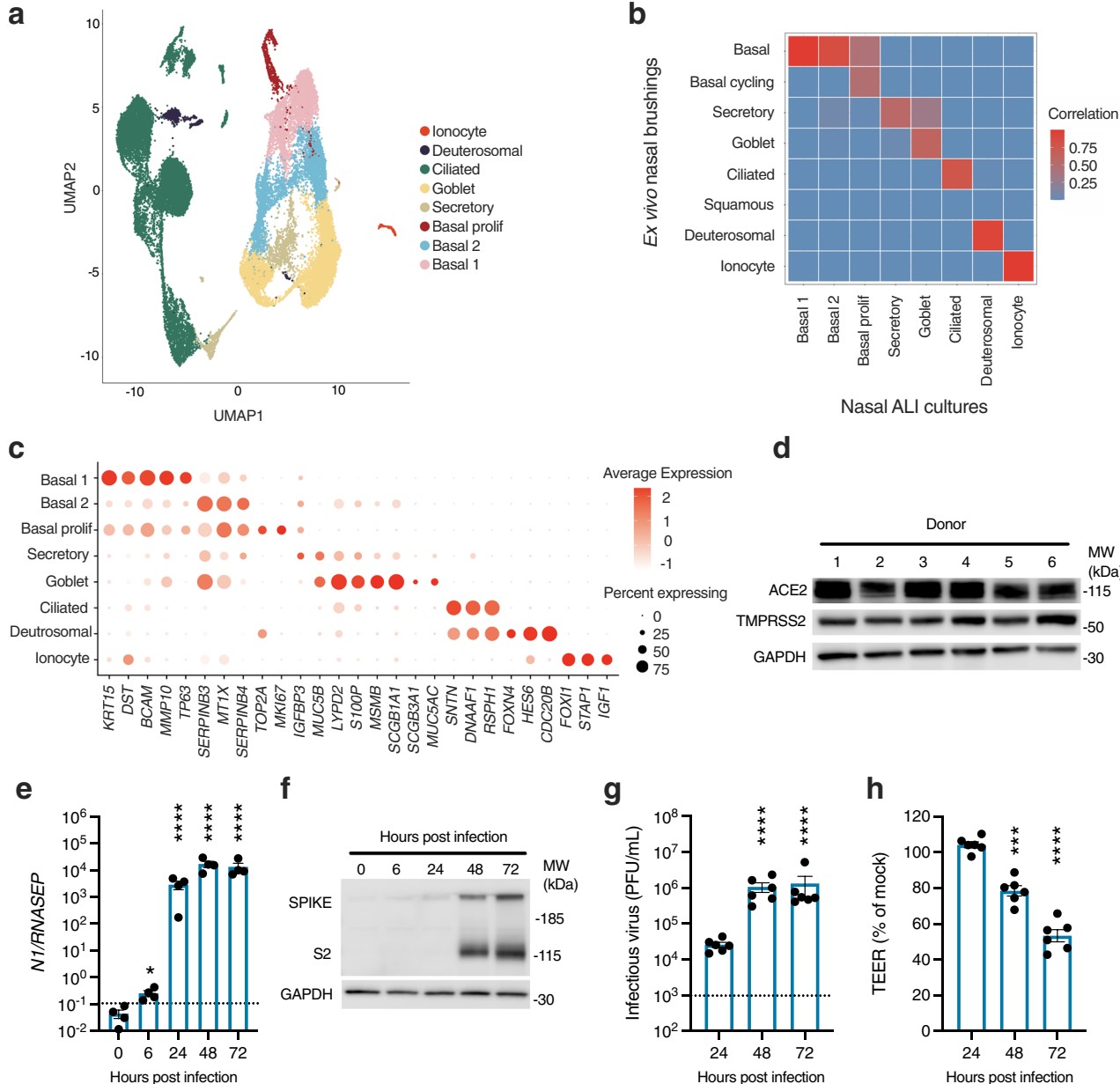

**Fig. 1 Robust SARS-CoV-2 infection in a primary differentiated nasal epithelial ALI culture model. a** UMAP visualisation of single-cell RNA sequencing (scRNA-seq) data from nasal ALI cultures (28,346 single-cell transcriptomes from two representative donors) showed six major cell types. **b** Correlation between the annotation from an external dataset of nasopharyngeal swabs and the assigned annotation of our scRNA-seq from nasal ALI culture following label transfer. **c** Dot plot demonstrating expression of key markers distinguishing cell types in annotated clusters, with intensity demonstrated by colour and size of the dot representing the proportion of cells expressing the marker. **d** Immunoblot demonstrating ACE2 and TMPRSS2 expression by donor, representative of $n = 3$ experiments. Nasal ALI cultures were infected with SARS-CoV-2 (MOI 0.1) and subjected to various modalities to analyse infection. Whole-cell lysates were prepared at the indicated times for RT-PCR analysis of expression of **e** SARS-CoV-2 nucleocapsid (*N1*) gene expression normalised to the housekeeper *RNASEP* (average of $n = 2$ repeat experiments in $n = 4$ donors, mean ± SEM; *$P = 0.0248$, ****$P < 0.0001$, ANOVA, two-sided, with Dunnett's post-test correction compared to 0 h). **f** Whole-cell lysates were prepared at the indicated times for immunoblot analysis of expression of SARS-CoV-2 spike (S) and cleaved S2 protein expression (representative of repeat experiments in $n = 4$ donors). **g** Release of infectious viral particles was measured by plaque assay of apical washings on permissive Vero E6 cells (average of repeat experiments in $n = 6$ donors, mean ± SEM; ****$P < 0.0001$, ANOVA, two-sided, with Dunnett's post-test correction compared to 24 h). Dotted line represents lower limit of detection. **h** Transepithelial resistance (TEER) measurements upon infection (expressed as % of mock-infected wells, $n = 6$ donors, mean ± SEM; ***$P = 0.0007$, ****$P < 0.0001$, ANOVA, two-sided, with Dunnett's post-test correction compared to 24 h). UMAP = Uniform Manifold Approximation and Projection, MOI = multiplicity of infection, PFU = plaque-forming units, MW = molecular weight, kDa = kilodalton, ACE2 = angiotensin-converting enzyme 2, TMPRSS2 = transmembrane serine protease 2, GAPDH = glyceraldehyde-3-phosphate dehydrogenase.

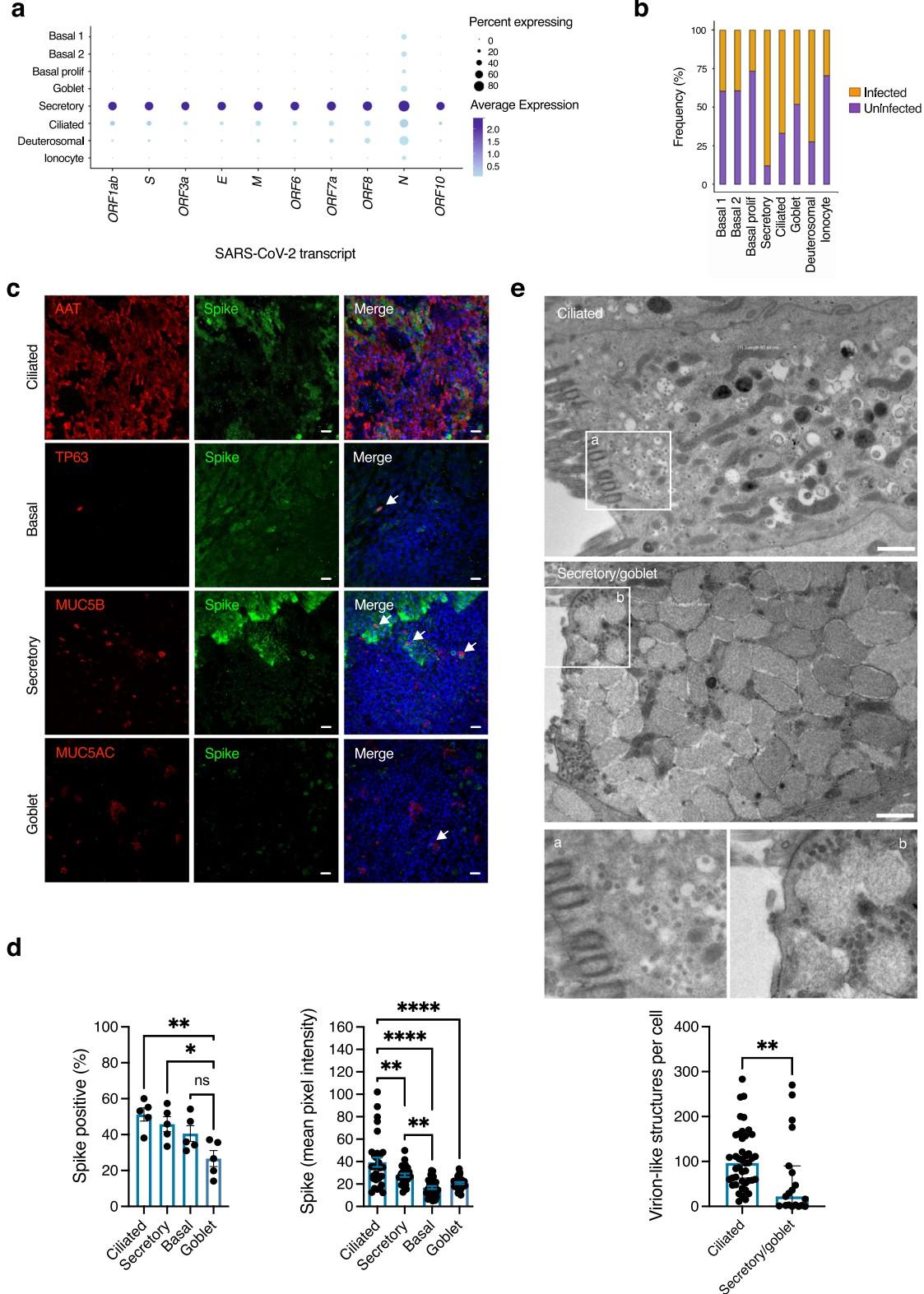

permissive to SARS-CoV-2[13,14], we first examined viral gene expression by scRNA-seq analysis at 24 hpi, selected to represent an early stage in the progress of infection. While all cell types expressed viral transcripts, there were notable differences both in the proportion of cells infected, and the relative abundance of different viral transcripts within these cells (Fig. 2a). Based on differential gene expression analysis between cell types (Wilcoxon rank sum test, one vs rest, $P < 0.05$), secretory and ciliated cells

expressed higher levels of viral transcripts than other cell types, with viral transcripts most abundant in secretory cells (Fig. 2a, Supplementary Data 2). Deuterosomal cells also expressed abundant viral transcripts but were a rare population within these cultures, possibly limiting the power of this analysis. Basal cells are also located away from the apical surface; physical inaccessibility to apically-applied virus at this time point might at least partially account for this observation. To investigate tropism

**Fig. 2 Broad tropism of SARS-CoV-2 for nasal cells.** Nasal ALI cultures were infected with SARS-CoV-2 (MOI 0.1) and analysed using different modalities to explore tropism. At 24 h post-infection (hpi), cell suspensions were prepared from two representative donors for single-cell RNA sequencing (scRNA-seq) and 28,346 individual transcriptomes passing QC were analysed. **a** Dot plot of scRNA-seq data showing magnitude (colour) and proportion (size) of cell types expressing viral transcripts. E = envelope; M = matrix; N = nucleocapsid; S = spike, ORF = open reading frame. **b** Relative proportion of infected cell types based on expression of any viral transcript. Separately, nasal ALI cultures were fixed at 48 hpi and subjected to immunofluorescence analysis. **c** Expression of viral S protein expression in ciliated (AAT), basal cells (TP63), secretory (MUC5B) and goblet (MUC5AC) cells (arrowed) shown in (**c**). Scale bars = 10 μm (representative of experiments in n = 5 donors). **d** Quantification of cell-type specific expression of viral S protein (Goblet vs Secretory *p = 0.0309, Goblet vs Ciliated **p = 0.0045) and S protein intensity (Basal vs Secretory **p = 0.0073, Secretory vs Ciliated **p = 0.0046, ****p < 0.0001) at 48 hpi (n = 5 donors, mean ± SEM; ns = non-significant, ANOVA, two-sided, with Sidak's post-test correction for multiple comparisons, indicated by lines). **e** Nasal ALI cultures were infected as above, fixed at 48 hpi for transmission electron micrograph (TEM) analysis of SARS-CoV-2 infected ciliated and secretory/goblet cells. Inserts **a**, **b** display virion-like structures in ciliated and secretory/goblet cells, respectively. Scale bars = 1 μm. Image analysis was undertaken to quantify virion-like structures as displayed in the bar plot (n = 3 donors, mean ± SEM **P = 0.0031, Mann–Whitney test, two-sided). MOI = multiplicity of infection, AAT = acetylated-alpha tubulin, tumour protein 63 = TP63, MUC = mucin.

further, we undertook immunofluorescence analysis of viral spike (S) protein expression at 48 hpi (Fig. 2c, d, see Supplementary Fig. 4 for background spike immunoreactivity in uninfected cells). This analysis showed broadly similar proportions of ciliated, secretory and basal cells expressing S protein, with significantly lower spike immunoreactivity in MUC5AC positive (goblet) cells (Fig. 2d). However, the mean pixel intensity of S protein was significantly greater in ciliated cells than in other cell types, and significantly increased in secretory cells compared to basal cells (Fig. 2d). To corroborate these findings, we undertook analysis of intracellular virion-like structures (VLSs) at 48 hpi by transmission electron microscopy (TEM, Fig. 2e), focusing on ciliated and secretory/goblet cells (the latter cell types were grouped for analysis as they could not be reliably distinguished based on morphology). Intracellular VLSs were observed in both ciliated and secretory/goblet cells, predominantly towards the apical surface (Fig. 2e). Consistent with immunofluorescence analysis of S protein intensity, there was a significant increase in the number of VLSs per cell in ciliated compared with secretory/goblet cells (Fig. 2e). Collectively, these data suggested that the virus is capable of entering, and replicating in, all major nasal cell types, but with quantitative differences in efficiency.

**Characterisation of individual nasal cell responses to SARS-CoV-2.** Published ex vivo single-cell transcriptomic analyses report that nasal cell types exhibit the baseline expression of an innate antiviral gene signature, in the absence of viral infection, characterised by several IFN-stimulated genes (ISGs)[10,11]. This signature correlated with *ACE2* expression, suggesting conditioned expression to reduce susceptibility. Based on these reports, we examined scRNA-seq data to characterise the innate antiviral response of nasal cells to SARS-CoV-2 infection at 24 hpi. In unexposed cells, ISG signature scores were generated using context-specific ISGs from a published IFN-treated nasal cell dataset[11] and compared using the Wilcoxon rank sum test. Gene set scores greater than zero suggested expression levels higher than background gene expression, and was the case for basal 1, secretory, goblet cells and ionocytes (Fig. 3a). The ISG signature was highest in secretory cells despite abundant expression of viral RNA in these cells upon exposure to SARS-CoV-2 (Fig. 2a, b), suggesting constitutive ISG expression may not be sufficient to protect against infection. Next, we distinguished cells in three experimental conditions: unexposed (mock-infected); SARS-CoV-2-exposed but uninfected (these 'bystander' cells would theoretically be exposed to IFNs and other paracrine signals, but not infected); and SARS-CoV-2-infected (as defined by detectable expression of SARS-CoV-2 transcripts). We undertook differential expression (DE) analysis between mock and bystander or infected cells, labelling ISGs derived from the same list of context-specific ISGs (Fig. 3b, Supplementary Data 3–4). There was

minimal transcriptional response to infection in bystander cells, including the absence of ISG induction, suggesting a lack of substantial paracrine IFN signalling at this timepoint in keeping with reports in other airway models[25,27,28]. Interestingly, *IFITM* genes (ISGs which have been paradoxically implicated in SARS-CoV-2 entry[33]) were downregulated in some bystander cell populations. There was also minimal evidence of ISG induction in SARS-CoV-2 infected cells, especially secretory, deuterosomal and goblet cells (Fig. 3b). In secretory and ciliated several ISGs relating to antigen processing or presentation were down-regulated upon infection (Fig. 3b). Basal cells expressed a modest number of ISGs upon infection, specifically genes of the *IFITM* family, *IFI27* and *IFI6* and the negative regulator of IFN-I signalling, *ISG15*. Consistent with this finding, gene-set enrichment analysis (GSEA) identified upregulation of IFN alpha/gamma responses in infected basal cell populations but not in other cell types (Fig. 3c). The transcriptional response of infected secretory and ciliated cells was characterised by widespread downregulation of expression, which may reflect viral co-optation of transcriptional machinery of host cells, but this effect was not uniform. In agreement with previous reports[34,35], genes related to oxidative phosphorylation were prominent amongst downregulated genes, as were antigen presentation pathways. Pathway analysis also predicted upregulation of NF-KB signalling in basal, secretory, goblet and ciliated cells, consistent with previous findings[27,36]. Using DoRoTHea to explore regulon activity in these populations confirmed higher predicted *NFKB2* activity but limited evidence of widespread activation of IFN-mediated signalling (Supplementary Fig. 5). Transcripts for IFN-I (*IFNB, IFNK, IFNA5*) and IFN-III (*IFNL1*) were not significantly differentially expressed and were detectable in only a small minority (~0.4%) of infected secretory cells (Supplementary Fig. 6). Whilst potentially consistent with the lack of paracrine signalling at this timepoint, this might also reflect transient expression and/or insensitivity of detection by scRNA-seq; a similar pattern was observed for other cytokines and chemokines (Supplementary Fig. 6). Overall, this analysis showed that despite evidence of NF-KB activation at 24 hpi, there was a minimal IFN response to SARS-CoV-2 in the cell types with the highest levels of infection, consistent with previous reports in non-nasal epithelial cells[25,27,28].

**Kinetics of innate IFN-I/IFN-IIIs response to SARS-CoV-2.** To investigate the kinetics of the IFN-I/III response, expression of IFN-I (*IFNA1* and *IFNB*) and IFN-III (*IFNL1*) was examined by RT-PCR at 6, 24, 48 and 72 hpi at the same MOI (0.1) as previous experiments. Induction of *IFNL1* and *IFNB* was low at 24 hpi, consistent with scRNA-seq findings, but increased significantly by 48 and 72 hpi (Fig. 4a). *IFNA1* was not induced, as observed in our scRNA-seq data. Compared to the timing of initiation of viral gene expression - which was detectable at 6 hpi and approached

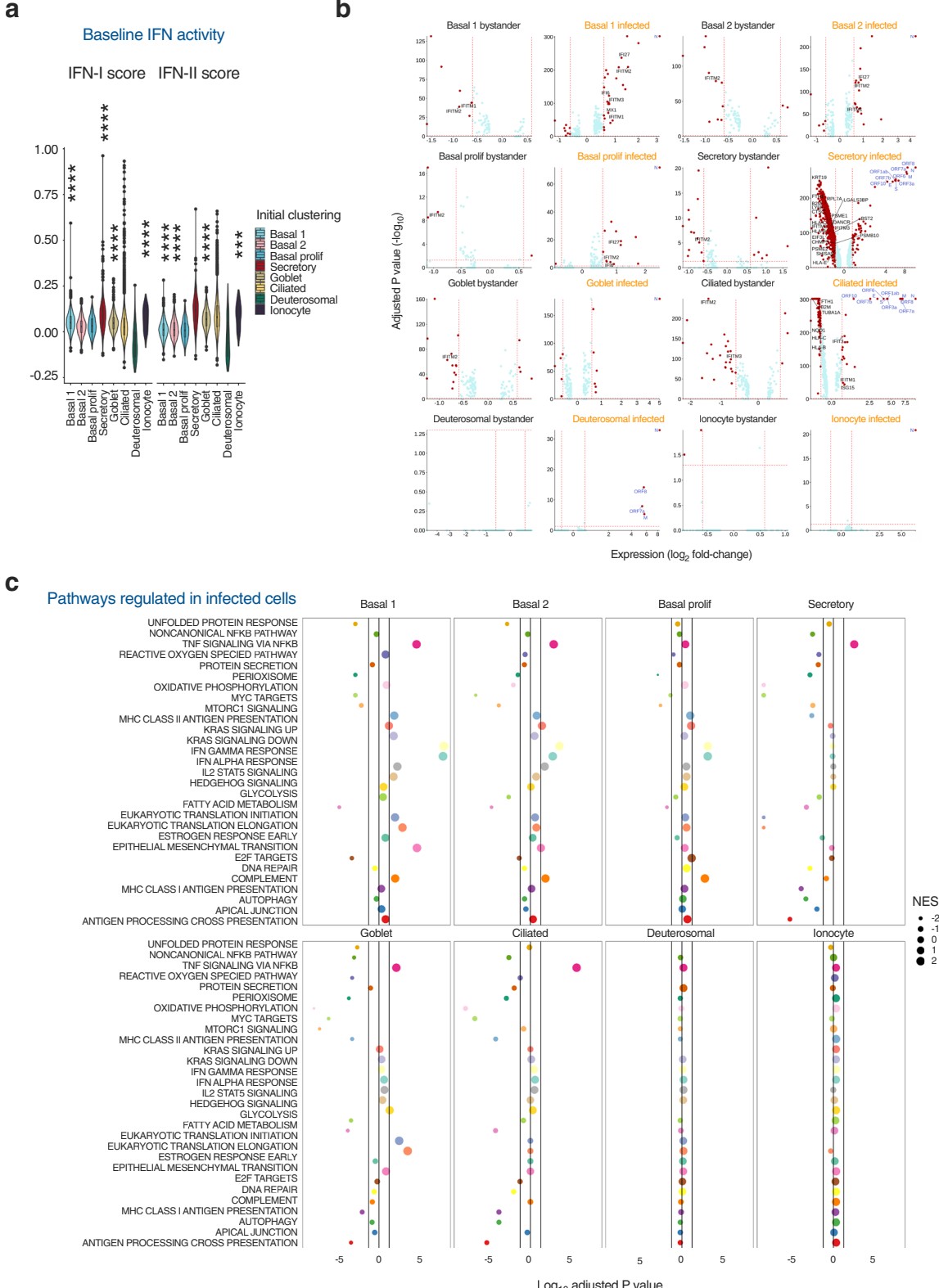

its maximum level by 24 hpi (Fig. 1e)—the induction of IFNs appeared delayed, as suggested by previous studies[12,14,36]. Infection was accompanied by progressive upregulation of proinflammatory cytokines such as *IL6, IL1B* and *TNF*, consistent with initiation of an NF-KB-dependent inflammatory response (Fig. 4b). To look for evidence of a paracrine response to IFN-I/III, we analysed expression of the ISGs *RSAD2* and *USP18* by

RT-PCR, as well as the expression of RSAD2, USP18, ISG15 and MX1 proteins by immunoblotting. There was an increase in ISG mRNA and protein expression at later times following the onset of *IFN* gene expression (Fig. 4c, d), potentially suggestive of paracrine JAK-STAT signalling. To explore early induction of IFNs in more detail, we compared the response to SARS-CoV-2 with other RNA respiratory viruses, influenza A virus (IAV) and

**Fig. 3 Characterisation of individual nasal cell transcriptional responses to SARS-CoV-2.** Nasal ALI cultures were infected with SARS-CoV-2 (MOI 0.1). At 24 h post-infection (hpi), cell suspensions were prepared from two representative donors for single-cell RNA sequencing (scRNA-seq) and 28,346 individual transcriptomes passing quality control (QC) were analysed. **a** Violin plot representing the composite interferon-stimulated gene (ISG) signature score that was defined based on a published nasal cell dataset from cells treated with IFN alpha and IFN gamma. Gene set scores greater than zero suggest expression levels higher than background gene expression. The bottom and the top of the boxes correspond to the 25th (Q1) and 75th (Q3) percentiles, and the internal band is the 50th percentile (median). The plot whiskers represent the 95% confidence intervals show down to the minimum (Q1−1.5*IQR) and up to the maximum (Q3 + 1.5*IQR) value. IQR = interquartile range. Outside points correspond to potential outliers. See Supplementary data 6 for exact values. Two-sided Wilcoxon rank sum testing was performed for each cell type vs all with Benjamini–Hochberg correction (***$P < 0.0008$, ****$P < 0.0001$). **b** Differential expression (DE) analysis by Wilcoxon rank sum test was undertaken to compare mock-infected cell transcriptomes with those from bystander cells (without detectable viral transcripts) and infected cells (with detectable viral transcripts) from the virus-exposed cultures. Volcano plots were generated with vertical lines marking ±1.5 fold change cut-offs (note $\log_2$ scale) and the horizontal line marking an adjusted $P$ value cut-off of 0.05 (<0.05 was considered statistically significant). Individual genes coloured as non-significant (light blue) and significant (red). Labels indicate viral transcripts (dark blue) and epithelial-cell specific ISGs (black). **c** Gene-set enrichment analysis was undertaken by ordering genes by fold change difference between mock-infected and infected cells by cluster (two-sided Wilcoxon rank sum statistical test with Bonferroni multiple testing correction). Vertical lines indicated adjusted $P$ value cut-off of 0.05. NES = normalised enrichment score, MOI = multiplicity of infection.

parainfluenza virus 3 (PIV3). In this experiment, significant induction of *IFNB* and *IFNL1* occurred in response to both PIV3 and IAV, but not SARS-CoV-2, at 24 hpi (Supplementary Fig. 7), and was accompanied by upregulation of ISGs *USP18* and *RSAD2*. Infection of cell lines at high MOI are reported to enhance the relatively inefficient IFN-I induction to SARS-CoV-2[25]. To confirm that the attenuated production of *IFNL1* and *IFNB* at early times was not dependent on MOI, we repeated SARS-CoV-2 infections at 20-fold higher MOI (2), alongside IAV (Fig. 4e), or a preparation of Sendai virus (SeV) containing a high proportion of immunostimulatory defective viral genomes[32] as a positive control (Supplementary Fig. 8). At 6 hpi, a time point at which *IFNB* and *IFNL1* were significantly induced by IAV, there was no detectable response to SARS-CoV-2 (Fig. 4e). Compatible observations were made with SeV (Supplementary Fig. 8). At 24 hpi, SARS-CoV-2 exposure led to no detectable induction of *IFNB* and significantly less *IFNL1* than IAV (Fig. 4e). This differential response was reflected in the robust expression of ISGs *RSAD2, USP18* and *ISG15* at 24 h post-inoculation with IAV but not SARS-CoV-2 (Fig. 4f). These observations recapitulated our previous RT-PCR and scRNA-seq data with a lower MOI, and are consistent with other reports[12,27,37]. Collectively, the results indicate that nasal epithelial cells express IFN-I/IIIs during SARS-CoV-2 infection, but that the response is delayed relative to viral replication.

**IFN-signalling dominates the nasal host response to SARS-CoV-2 at the protein level.** To validate and extend these findings, we undertook an unbiased assessment of the host response to SARS-CoV-2 infection by proteomics analysis. Whole-cell lysates were prepared from SARS-CoV-2 and mock-infected nasal ALI cultures from six donors at 72 hpi. Lysates were analysed by quantitative mass spectrometry (quality control data in Supplementary Fig. 9). Overall, this analysis detected the differential expression (DE) of 180 proteins including viral proteins such as S, M, N, ORF1AB, ORF3A and ORF8 (Fig. 5a, Supplementary Data 5). The most highly increased host protein was Sorting Nexin 33 (SNX33), an endosomal protein that has not yet been implicated in the life cycle of SARS-CoV-2. Notably, other SNX proteins (e.g. SNX17 and SNX2) are involved in viral trafficking[38,39]. Infected and uninfected cells clustered together by principal component analysis (Fig. 5b). Inspection of the DE proteins confirmed a robust host innate immune response, dominated by ISG products (Fig. 5a). Functional annotation identified an enrichment of antiviral response and especially IFN-I signalling pathways (Fig. 5c, Supplementary Table 2). These data are consistent with our earlier findings and contrary to prior reports in cell lines or human bronchial/tracheal epithelial

cultures, where a robust endogenous IFN-I/III response to SARS-CoV-2 was not detected[26–28]. Key antiviral ISG proteins identified included IFIT1-3, MX1-2, and the OAS cluster (OAS1-3), the latter associated with genetic susceptibility to severe COVID-19[19] (Fig. 5c). Significantly downregulated pathways were also identified, including TRIF-dependent toll-like receptor signalling, as well as RNA polymerase II transcription and endosomal transport (Supplementary Table 3). This implied viral subversion of critical host functions, including host gene transcription, protein trafficking and viral sensing. Proteins involved in the maintenance of epithelial tight junctions were also downregulated, consistent with the loss of barrier integrity observed in earlier experiments (Fig. 1h).

**Antiviral activity of IFN-I/III towards SARS-CoV-2 infection.** Given the prominence of the IFN-I/III response in the proteome of SARS-CoV-2-infected cells at later times post-infection, a key question was whether this IFN-I/III response had any impact on SARS-CoV-2 replication. To address this question, nasal ALI cultures were treated with the JAK inhibitor ruxolitinib (RUX). RUX antagonises signalling downstream of IFNAR and IFNLR, owing to the involvement of JAK1 in both signalling pathways. We reasoned that blocking paracrine IFN-I/III signalling would establish its impact, if any, on SARS-CoV-2 replication. Cells were treated with 10 μM RUX (a dose optimised in prior experiments[40]) or vehicle control (DMSO) in the basal medium for 24 h prior to infection. Nasal cultures were infected at the apical surface (MOI 0.1), inhibitors were refreshed every 24 h and infection was monitored up to 96 hpi. Lysates were prepared and analysed by RT-PCR and immunoblot. RUX treatment abolished expression of ISGs *USP18, RSAD2* and *ISG15* at the mRNA and protein level (Fig. 6a, b), indicating that ISG induction was dependent on paracrine IFN-I/III signalling, as previously suggested (Fig. 4d). By 96 hpi, approximately 24 h after ISGs were reliably detected at the protein level (Figs. 4d, 5a–c), blockade of this endogenous IFN response by RUX led to a significant increase in both viral gene expression, assessed by RT-PCR (*N* gene) and immunoblot (S/S2 protein, Fig. 6b–d), and apical release of infectious virus measured by plaque assay (Fig. 6e). These data provided further evidence that SARS-CoV-2 triggered an endogenous paracrine IFN-I/III response in nasal cells, which once established began to impact SARS-CoV-2 replication.

An important follow-up question was whether nasal cells could mount an antiviral state to SARS-CoV-2, providing IFN-I/III was delivered in a timely fashion. To address this, nasal ALI cultures were pre-treated with exogenous IFNβ (1000 IU/mL) or IFNλ1 (100 ng/mL) for 16 h to induce an antiviral state, subsequently infected with SARS-CoV-2 at MOI 0.01 and examined at 48 hpi.

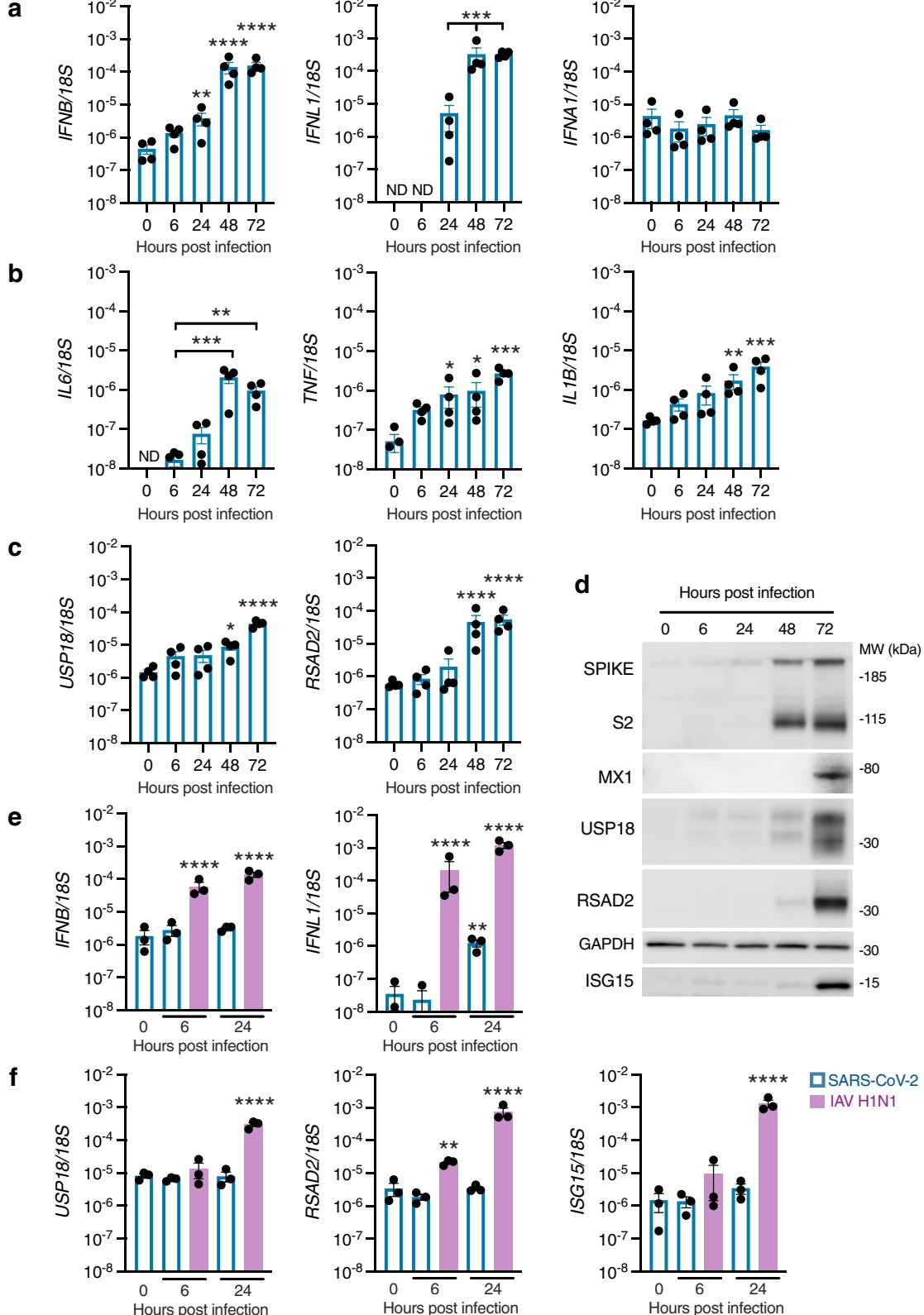

Analysis of infection by immunoblotting of whole-cell lysates for spike (S/S2) protein expression or plaque assay of apical washes demonstrated a significant reduction in infection with either IFNβ or IFNλ1 pre-treatment (Fig. 7a, b). This was accompanied by robust induction of antiviral ISG products (Fig. 7a), and preservation of barrier integrity (Supplementary Fig. 10). It is worth noting that the ISG expression induced in response to

recombinant IFN-I/III at 48 hpi was substantially greater than that induced by endogenous IFN-I/III production (Fig. 7a). Thus exogenous IFN-I/III was capable of inducing in the nasal epithelium an antiviral state that potently inhibited SARS-CoV-2 infection, providing it was delivered (a) prior to infection, and (b) at sufficient concentration. This IFN-sensitivity of SARS-CoV-2 contrasts with the relative resistance of SARS-CoV[21].

**Fig. 4 Delayed induction of IFN-I/III signalling in SARS-CoV-2-infected nasal ALI cultures.** Nasal ALI cultures were infected with SARS-CoV-2 (MOI 0.1). Whole-cell lysates were prepared at the indicated times for RT-PCR analysis of expression of **a** *IFNB* (**$p = 0.0081$, ****$p < 0.0001$), *IFNL1* (***$p < 0.0006$) and *IFNA1* **b** *IL6* (*$p = 0.001$, ***$p = 0.0003$), *TNF* (0 h vs 24 h *$p = 0.0182$, 0 h vs 48 h **$p = 0.0124$, ***$p = 0.0002$) and *IL1B* (**$p = 0.0031$, ***$p = 0.0001$) and **c** *USP18* (*$p = 0.0102$, ****$p < 0.0001$), and *RSAD2* (****$p < 0.0001$), (average of $n = 2$ repeat experiments in $n = 4$ donors, mean ± SEM; ANOVA, two-sided, with Dunnett's post-test correction compared to 6 h [**b**, *IL6*], 24 h [**a**, *IFNL1*] or 0 h [all others]). ND, Not detected. **d** Whole-cell lysates were prepared at the indicated times for immunoblot analysis of Spike/cleaved S2, MX1, USP18, RSAD2 and ISG15 expression (representative of experiments in $n = 4$ donors). Nasal ALI cultures were infected with SARS-CoV-2 or influenza A virus (IAV H1N1, purple bars) at MOI 2. Whole-cell lysates were prepared at the indicated times for RT-PCR analysis of expression of **e** *IFNB*, *IFNL1* (**$P < 0.0056$, ****$P < 0.0001$, all compared to 0 h) and **f** the ISGs *USP18*, *RSAD2* and *ISG15* (**$P < 0.0010$, ****$P < 0.0001$, all compared to 0h) ($n = 3$ donors, mean ± SEM; ANOVA, two-sided, with Dunnett's post-test correction). MOI = multiplicity of infection, MW = molecular weight, kDa = kilodalton.

These data suggest that mucosal delivery of IFNβ or IFNλ1 is a potential therapeutic strategy for SARS-CoV-2. In clinical practice, IFNs are unlikely to be used prior to infection, unless this is part of a prophylactic regimen. To examine the effectiveness of exogenously applied IFNs once SARS-CoV-2 infection is underway, infected cells were treated with IFNβ or IFNλ1 at 6 or 24 hpi and examined for S/S2 protein expression by immunoblot (Fig. 7c, d) and release of infectious virus by plaque assay (Fig. 7e). In this experiment, IFNβ and IFNλ1 treatment at 6 hpi continued to impact on SARS-CoV-2 infection, whereas addition after 24 hpi had minimal effect. Interestingly, ISG induction was still observed in response to IFN treatment at 24 hpi, albeit at reduced magnitude in the case of IFNβ (Fig. 7c, d). These data suggest that SARS-CoV-2 may impair, but does not abolish, JAK-STAT signalling in infected cells, implying that recombinant IFNs may be therapeutically applicable to established SARS-CoV-2 infection, as recently shown in animal models[9] and in early phase clinical trials[41].

## Discussion

We report the most comprehensive characterisation of the human nasal epithelial response to experimental SARS-CoV-2 infection to date, observing a response dominated at later stages by IFN-I/IIIs and their downstream ISG products. This response partially contained SARS-CoV-2 at later times post-infection, while recombinant IFN-I/III treatment potently blocked SARS-CoV-2 replication, suggesting that mucosal delivery of IFNs could be a promising strategy for post-exposure prophylaxis.

The nasal mucosa is likely to be a main point of entry of SARS-CoV-2. Prior single-cell transcriptomic studies implied an abundance of target cells in the nasal mucosa and further suggested that they may be poised to mount an antiviral response[10]. Yet few studies to date have characterised SARS-CoV-2 replication in primary human differentiated nasal cells[12–14], while we analyse the host-virus interaction comprehensively, at single-cell resolution and utilising proteomics. Our findings indicate that the host response to SARS-CoV-2 in nasal epithelium is dominated by paracrine IFN-I/III signalling, albeit this response is kinetically delayed. These data contrast with initial reports that SARS-CoV-2 did not induce a robust IFN response in airway epithelial cells[26–28], but are consistent with emerging evidence of IFN-I/III induction in nasal swabs from patients with COVID-19[15,42–44] and with more recent findings in lung airway models[36,43,45–47]. Blockade of the endogenous IFN response had an impact on SARS-CoV-2 infection at later stages post-infection, once the IFN response was established, underscoring the delayed kinetic but also emphasising its functional relevance. While the impact of endogenous IFN-I/III signalling upon SARS-CoV-2 replication has not to our knowledge been investigated in nasal cell models, our data are consistent with recent findings in some[43,47], but not all epithelial model systems[27,45]. Our experiments with IAV, PIV3 and SeV—viruses which induced the robust early expression of IFN-I/III, in line with the previous studies[12,27,37]—

confirm that this delay was not due to an intrinsic property of nasal epithelial cells. The expression of IFN evasion proteins[37,48], the sequestration of viral replication machinery within cytosolic vesicles[49], as well as global reductions in host mRNA content[50] and translational shutdown[51–53] induced by SARS-CoV-2 presumably underlie its capacity to subvert early IFN induction in infected cells. Consistent with this, there was evidence of down-regulation of immune pathways including TLR signalling in the proteome of infected cells. However, an important question is what molecular patterns are responsible for IFN-I/III induction at later times. Recent evidence implicates MDA5 as a major sensor of SARS-CoV-2 RNA in epithelial cells[45–47], while other findings suggest a contribution from virus-mediated damage occurring after several days of infection[54]. It will be important to address the relative contribution of host damage-associated molecular patterns versus viral pathogen-associated molecular patterns (e.g. defective viral genomes) accumulating during replication.

IFN-I/III signalling is plausibly implicated in protection against life-threatening COVID-19[18–20]. Consistent with this, circulating immune cells of patients with severe COVID-19 exhibit impaired ISG responses[55–57]. However, whether the local airway IFN response in the early stages of infection shapes the subsequent clinical outcome of COVID-19 remains to be conclusively determined. A compelling recent scRNA-seq study reported that patients going on to develop severe disease exhibited a muted ISG response in the nasal airway, in contrast to those with milder disease[15], and is supported by independent findings of attenuated nasal ISG induction in patients with autoantibodies to IFN-I[58], who are prone to more severe disease[20]. Additional strands of evidence suggest a potential link between airway IFN-I/III competence and clinical outcome in COVID-19. Age remains the strongest risk factor for poor outcome in COVID-19, and the efficiency of IFN-I/III induction is known to decline with advancing age[59], and appears to be greater in the nasal airways of children than adults infected with SARS-CoV-2[60]. Other relevant environmental influences, such as exposure to cigarette smoke or other viral infections, are also reported to perturb IFN-I/III responses of airway cells in ways that may be relevant to COVID-19 pathogenesis[43,61].

The main limitation of our data in this nasal epithelial culture system is that it did not account for professional immune cells present in the nasal mucosa, for example, plasmacytoid dendritic cells[62], which are capable of more rapidly mounting an IFN-I/III response to SARS-CoV-2[63], potentially tipping the scales in favour of the host[64]. We studied cells derived from adult donors, however, it is possible that nasal cells from paediatric donors, who are naturally less susceptible to severe COVID-19, may behave differently in terms of their reduced permissiveness to SARS-CoV-2 and/or the greater efficiency of their innate IFN response[60,65]. Furthermore, SARS-CoV-2 variants with mutations in the spike gene have emerged worldwide whilst we were undertaking the experiments described here; these variants may impact viral replication and/or host immunity, and should be included in future studies.

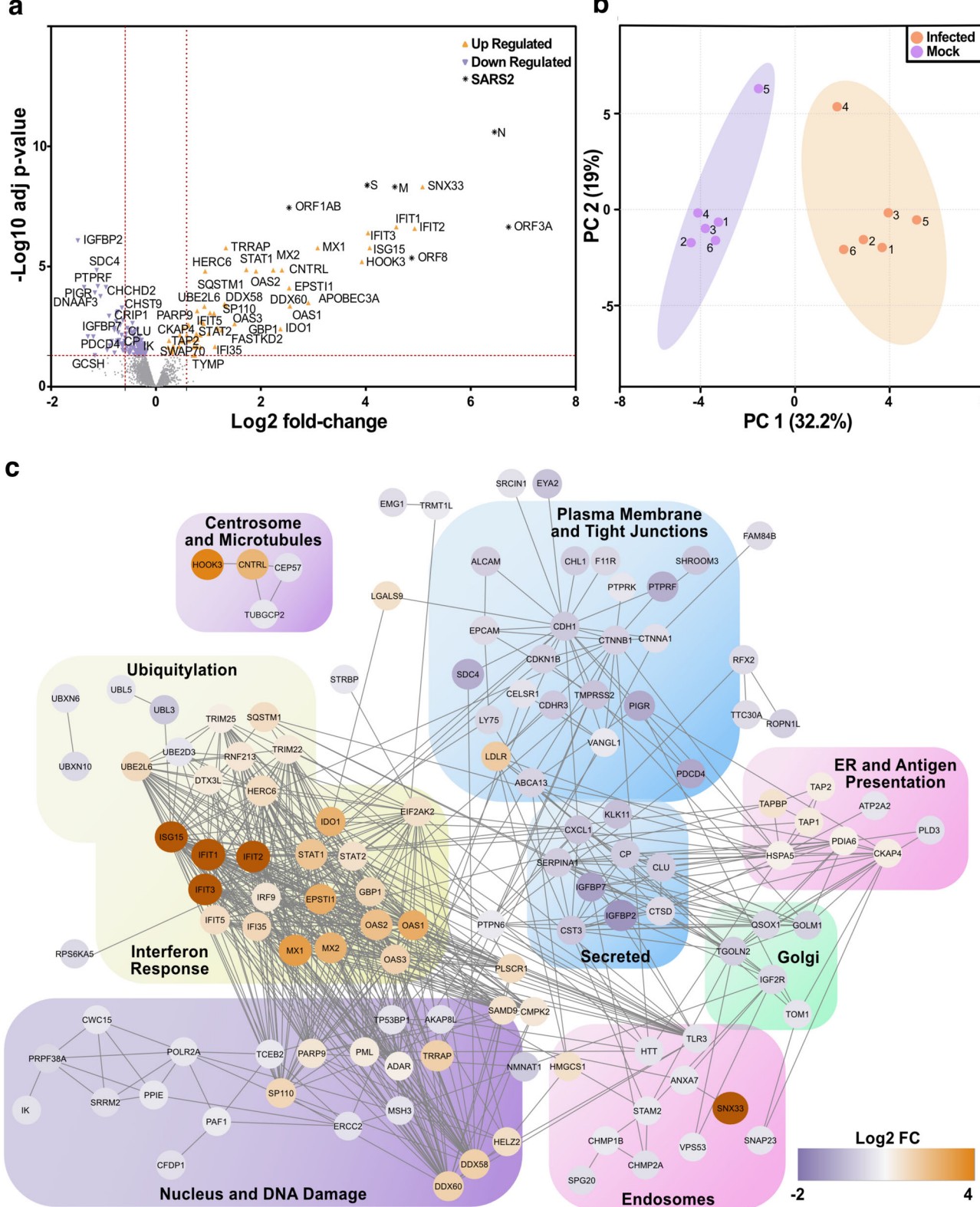

**Fig. 5 An ISG response dominates the proteome of SARS-CoV-2 infected nasal ALI cultures.** Differential proteomic profiling of SARS-CoV-2-infected nasal ALI cultures. Mass spectrometry-based proteomics was carried out on whole-cell lysates prepared at 72 h post infection (hpi; $n = 6$ donors per condition). The exact adjusted (adj) $P$ values can be found in Supplementary dataset 5. **a** Volcano plot illustrating 180 differentially expressed proteins with increased (orange points) and decreased (purple points) expression in infected as compared to mock-infected samples. Dotted red lines indicate those proteins with a fold change of >1.5 and adjusted $p$ values <0.05. **b** Principal component (PC) analysis of the whole proteome dataset. **c** Functional annotation network of differentially expressed proteins to indicate those proteins with a fold change (FC; Log2).

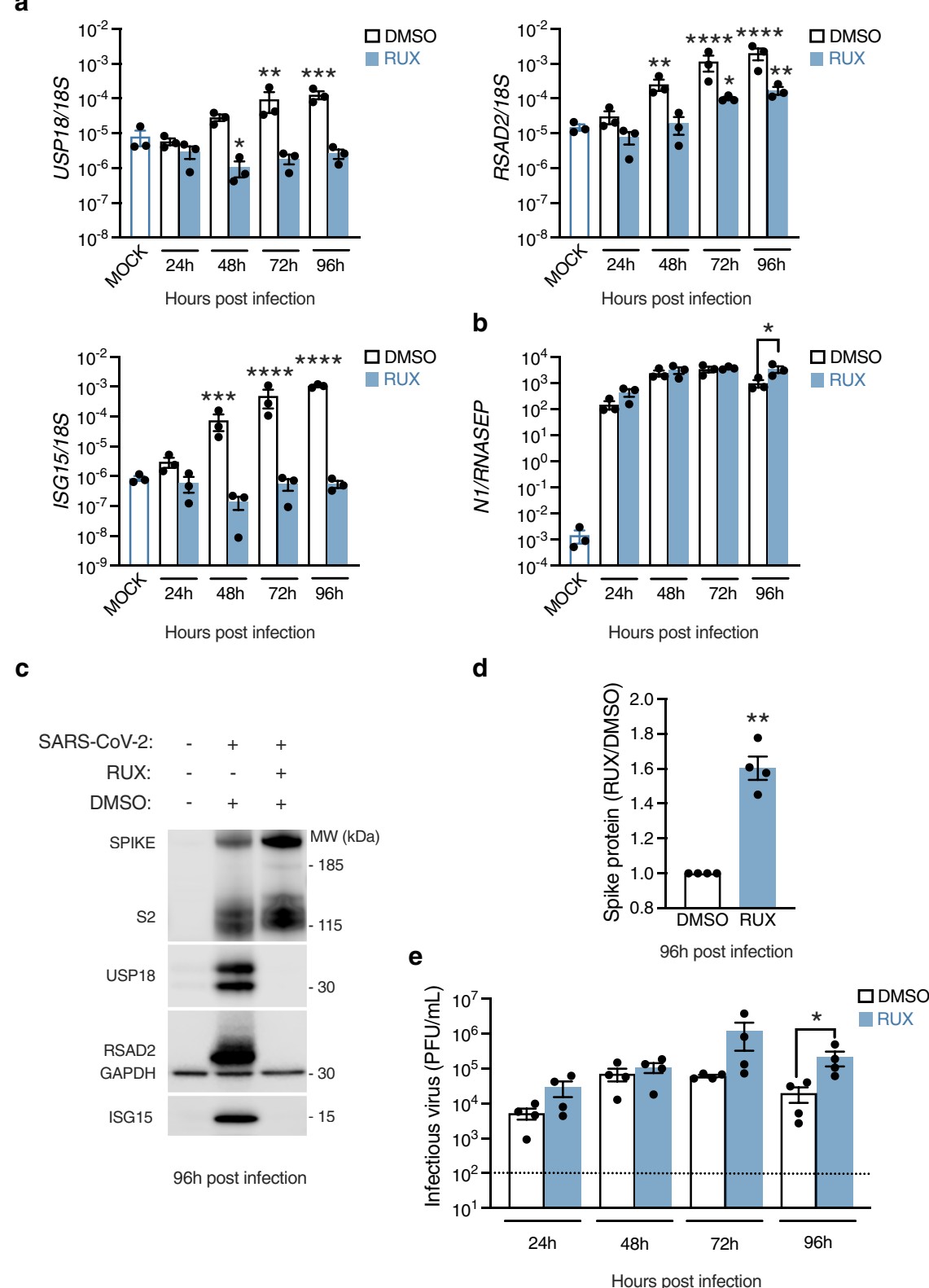

Nevertheless, our data, employing a variety of complementary methods, indicate that SARS-CoV-2 has a relatively broad tropism for nasal epithelial cells, confirming suggestions from prior scRNA-seq studies[8,10], other in vitro studies of primary nasal[14] and tracheobronchial cells[24,36], and importantly recent scRNA-seq studies of nasal samples from COVID-19 patients[15]. We also identify tropism for the rare deuterosomal cell, marked by expression of FOXN4, as recently reported[36,61,66]. Our findings contrast with the results of Hou and colleagues, who reported exclusive tropism of SARS-CoV-2 for ciliated cells in the nasal airway[13]. It is not immediately clear how to reconcile these findings, given that secretory cells express relevant entry receptors[10,13] and have been identified as a major infected cell type in infected patients[66]. Hou and colleagues used a fluorescent

**Fig. 6 Impact of endogenous IFN-I/III signalling on SARS-CoV-2 infection.** Nasal ALI cultures treated with ruxolitinib (RUX, 10 μM) or Dimethyl sulfoxide (DMSO) vehicle for 24 h prior to infection (MOI 0.1). Whole-cell lysates were prepared at the indicated times for RT-PCR analysis of expression of **a** the ISGs USP18 (*p = 0.0133, **p = 0.0051, ***p = 0.0006), RSAD2 (*p = 0.0333, mock vs DMSO 48 hpi **p = 0.0019, mock vs RUX 96 hpi **p = 0.0061, ****P < 0.0001) and ISG15 (***p = 0. 0006, ****P < 0.0001) (n = 3 donors, mean ± SEM; ANOVA, two-sided, with Dunnett's post-test correction, all compared to mock-infected cells) or **b** viral N mRNA (n = 3 donors, mean ± SEM; *P = 0.0346 ANOVA, two-sided, with Sidak's post-test correction compared to DMSO control). **c** Whole-cell lysates were prepared at 96 hpi for immunoblot analysis of viral Spike (S)/cleaved S2 protein and host RSAD2, USP18 and ISG15 protein expression (representative blot shown of experiments in n = 4 donors). **d** Densitometry analysis of S+S2 protein intensity relative to GAPDH, normalised to the DMSO control (data from **c**, n = 4 donors, mean ± SEM; **P = 0.003, one-sample t test, two-sided). **e** Plaque assay of apical washes collected at the times indicated showing a significant increase in infectious particle release at 96 h post infection (hpi) (same experimental conditions as (**c**, **d**); n = 4 donors, mean ± SEM; *P = 0.0147, ANOVA, two-sided, with Sidak's post-test correction compared to DMSO control). Dotted line indicates lower limit of assay detection. MOI = multiplicity of infection, MW = molecular weight, kDa = kilodalton, PFU = plaque-forming units.

reporter virus, the tropism of which might have been slightly narrower than clinical isolates. It is also worth noting that while we show that all cell types contained SARS-CoV-2 protein, there was a significant reduction in the proportion of goblet cells expressing spike protein, and the intensity of spike immunodetection was significantly greater in ciliated and secretory cells than basal or goblet cells. Ciliated cells also contained more virion-like structures per cell. Collectively, this implies that although all cell types are permissive to SARS-CoV-2 entry, there may also be quantitative differences in the overall efficiency of viral replication in different cell types. Hou and colleagues previously hypothesised that post-entry factors such as intrinsic antiviral immunity might dictate permissiveness. As discussed, we found limited evidence to support such a correlation, since while virtually all nasal epithelial cells demonstrated a baseline ISG signature— consistent with ex vivo nasal biopsy data[10]—this was apparently insufficient to mediate resistance to SARS-CoV-2, at least at the time point analysed. However, it remains possible that cell-type specific differences in the efficiency of induction of the IFN response (for example in basal cells) might contribute to more subtle variation in permissiveness.

The differential response of basal cell types to SARS-CoV-2 at 24 hpi identified by our scRNA-seq analysis is notable. Basal cells are the stem/progenitor cell population of the airway[67]. Recent data indicate an emerging function for these cells as sentinels of the airway inflammatory response[68]. For example, basal cells detect apoptotic cells in the context of viral inflammation[69], retaining memory of prior immune exposure[70]. More generally, stem/progenitor cell types exhibit enhanced intrinsic antiviral immunity[71]. Future studies should consider mechanism(s) governing the seemingly distinct early antiviral response of nasal airway basal cells to SARS-CoV-2, and its functional relevance.

Importantly, from a clinical perspective, the observation that IFN-I/III treatment prevented SARS-CoV-2 infection in vitro indicates that chemoprophylaxis with IFN-I/III may have therapeutic value. This approach has already been tested in a small clinical trial in China (although the absence of a control group makes it impossible to judge the efficacy of this approach[72]). Immunisation is the most tractable approach for large-scale primary prevention of COVID-19. However, owing to incomplete vaccine coverage, and reduced vaccine effectiveness in immunocompromised populations or against mildly symptomatic or asymptomatic infection, allied to the emergence of variants that may compromise vaccine efficacy, there will likely continue to be a need for targeted chemoprophylactic therapies to prevent transmission in specific circumstances. These include post-exposure prophylaxis of contacts—to avoid the need for self-isolation—as well as pre-exposure prophylaxis for certain high-risk encounters (e.g. in healthcare settings or prior to long-distance travel). Our data suggest that nasal application of IFNβ or IFNλ1 might have important applications in this setting and argue for urgent clinical assessment of this approach. In terms of

the therapeutic efficacy of mucosally-administered IFNβ in patients with established COVID-19[41], our findings suggest that early administration may be a key factor determining clinical efficacy. Furthermore, studies in animal models indicate that administration of IFNβ or IFNλ1 later in the disease course may have deleterious effects on viral inflammation and/or airway cell regeneration[73–75], suggesting the existence of a relatively narrow therapeutic window of opportunity.

In summary, we have shown that SARS-CoV-2 exhibits broad tropism for nasal epithelial cells, but with preferential infection of ciliated and secretory cell types. Nasal cells mount a robust innate antiviral response to SARS-CoV-2 dominated by paracrine IFN-I/III signalling, which is delayed in onset relative to viral replication, but which is nevertheless capable of exerting partial control at later times post-infection. Upon exposure to exogenous IFN-I/III treatment, these cells adopt a profound antiviral state, highlighting a potential clinical role for recombinant IFNβ or IFNλ1 in chemoprophylaxis and/or therapy of COVID-19.

## Methods

**Adult nasal airway epithelial cell culture at air–liquid interface.** Adult primary human nasal airway epithelial cells were derived from excess clinical material obtained during routine nasal surgical procedures[29]. Ethical approval for sample collection was provided via the Newcastle and North Tyne Research Ethics Committee (Reference 17/NE/0361) and written informed consent was provided prior to sample collection. Participants were not compensated for their sample donation. Tissue shaved from the superficial surface of the sample was chopped into ~2 mm² pieces and added to RPMI-1640 basal medium containing 0.1% protease (Sigma-Aldrich, UK) and incubated overnight with gentle agitation at 4 °C. All large pieces of tissue were discarded, and residual protease was neutralized with 5% FCS. The preparation was centrifuged (200 × g; 7 min) and the pellet resuspended in PneumaCult-Ex Plus expansion medium (Stemcell Technologies), then seeded onto 25 cm² tissue culture flasks pre-coated with 30 μg/mL Type I collagen (PureCol, Advanced BioMatrix). Flasks were incubated in a humidified atmosphere containing 5% CO₂ at 37 °C, with medium replaced every 48 h. Cells were trypsinised at 60–80% confluence and cryopreserved for future use. Upon thawing, cells were grown through an additional expansion phase, then transferred in Ex Plus medium onto collagen-coated 6.5 mm polyester transwell membranes with 0.4 μm pore size (Corning) at a density of 150,000 cells/cm². When cells were fully confluent, apical medium was removed and basolateral medium was switched to PneumaCult-ALI-S (Stemcell Technologies). Cells were maintained at air–liquid interface until fully differentiated. Barrier integrity of ALI cultures was monitored by measuring trans-epithelial electrical resistance (TEER, EVOM 2, World Precision Instruments). ALI cultures were validated for use in experiments based on microscopic appearance of appropriate ciliated morphology and TEER > 500 Ω * cm². The sex and age of donors are included in Supplementary Table 4.

**Viruses, cytokines and inhibitors.** A clinical isolate of SARS-CoV-2 (BetaCoV/ England/2/2020) was obtained from Public Health England (PHE). This was isolated from a patient in January 2020 and thus represents an early strain of SARS-CoV-2, not known to be affected by variants of concern. The initial stock was propagated once in Vero E6 cells. The same viral stock was used for all experiments. As SARS-CoV-2 is a Hazard Group 3 pathogen (Advisory Committee on Dangerous Pathogens, UK), all infection experiments were performed in a dedicated Containment Level 3 (CL3) facility by trained personnel. Sendai virus (Cantell strain) and parainfluenza virus 3 (PIV3) was obtained from Richard Randall (St Andrew's University). Influenza A virus (A/PR8/1934/H1N1) was propagated and titred on MDCK cells. For nasal ALI infections, apical poles were gently washed once with warm Dulbecco's modified Eagle's medium (DMEM;

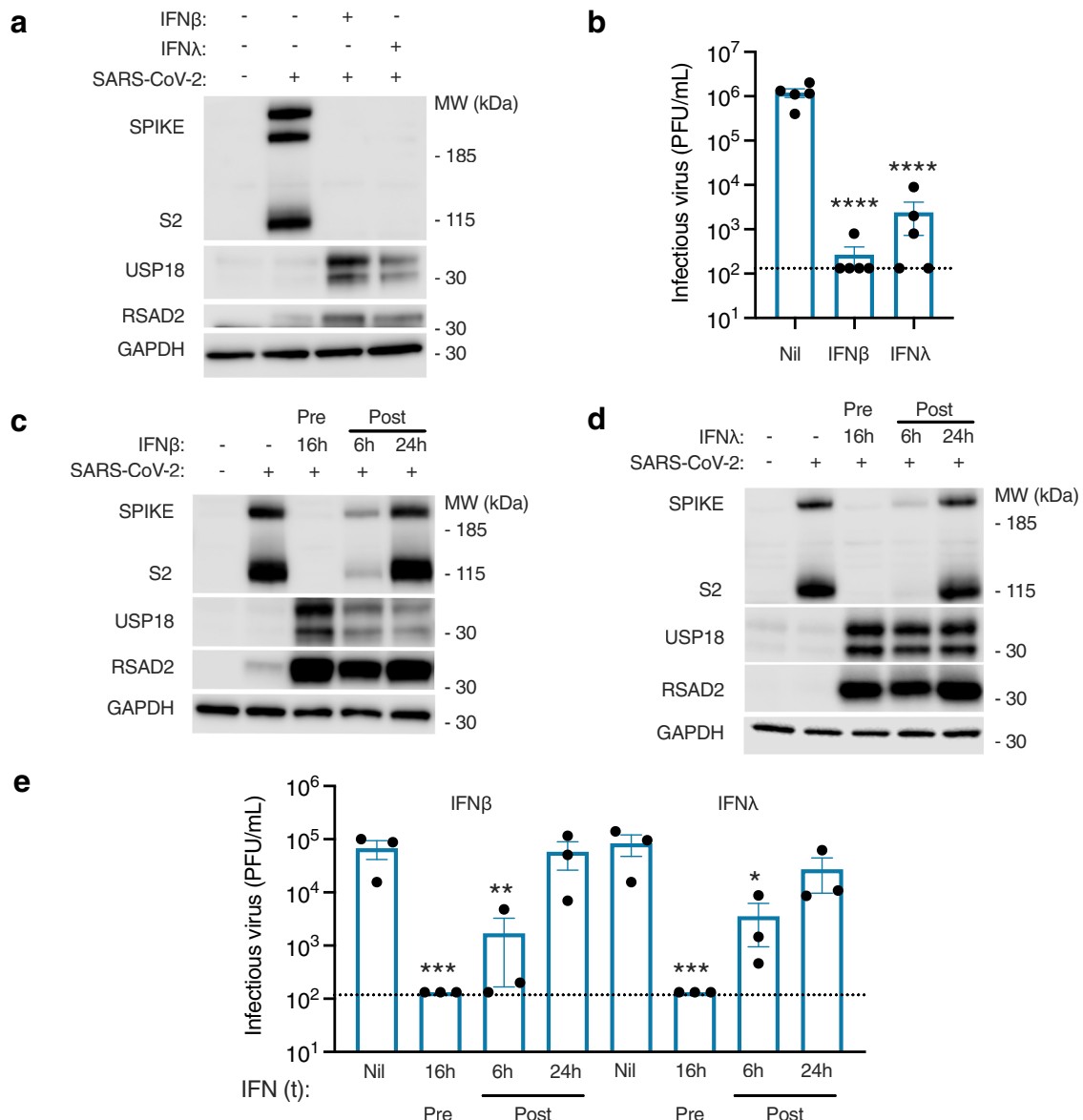

**Fig. 7 Exogenous IFN-I/III treatment controls SARS-CoV-2 replication.** Nasal ALI cultures were pre-treated for 16 h with IFNβ (1000 IU/mL) or IFNλ1 (100 ng/mL) prior to infection (MOI 0.01). **a** Immunoblot of whole-cell lysates prepared from nasal ALI cultures at 48 h post infection (hpi) (representative of experiments in $n = 4$ donors. **b** Plaque assay of apical washes showing significant reduction in infectious particle release at 48 hpi if pre-treated with IFNβ (1000 IU/mL) or IFNλ1 (100 ng/mL) (same experimental conditions as **a**; $n = 5$ donors, mean ± SEM; ****$P < 0.0001$, ANOVA, two-sided, with Dunnett's post-test correction compared to untreated control). **c**, **d** Immunoblot of whole-cell lysates prepared at 48 hpi. Nasal ALI cultures were either pre-treated (Pre) with IFNβ (1000 IU/mL, **c**) or IFNλ1 (100 ng/mL, **d**) for 16 h prior to infection with SARS-CoV-2, or IFN treatment was applied at 6 or 24 hpi (Post). Results representative of experiments in $n = 3$ donors. **e** Plaque assay on apical washes collected at 48 hpi from experiments in (**c**, **d**) ($n = 3$ donors, mean ± SEM; *$P = 0.0202$, **$P = 0.015$, ***$P = 0.001$, ANOVA, two-sided, with Dunnett's post-test correction compared to untreated control). Dotted line indicates lower limit of assay detection. MOI = multiplicity of infection, MW = molecular weight, kDa = kilodalton, PFU = plaque-forming units, Nil = not treatment, t = treatment.

Gibco, USA) and then infected with 60 μL dilution of virus in DMEM, at a MOI between 2 and 0.01 plaque-forming units per cell for 2 h, when the virus-containing medium was removed. DMEM was used as inoculum for mock infection. Apical washes (in warm phosphate-buffered saline) were collected at different time points and stored at −80 °C for plaque assays. Plaque assays were undertaken in Vero E6 cells using a 1.2% (w/v) microcrystalline cellulose overlay (Sigma-Aldrich). Cytokines/inhibitors were used at the following concentrations: human recombinant IFNβ1 (1000 ng/mL; Avonex, NDC 59627-002-06, Biogen Inc, USA); IFNλ1 (100 ng/mL; 1598-IL-025, R&D Systems, USA); and ruxolitinib (10 μM; S1378, Calbiochem, USA) alongside the appropriate dilution of DMSO vehicle. Treatment was applied through basolateral poles.

**Single cell RNA sequencing sample processing**. For the droplet-encapsulation scRNA-seq experiments, ALI cultures were washed with PBS and then incubated

with 1x Trypsin-EDTA (ThermoFisher Scientific, USA) for 10 min before the cells were diluted with DMEM and counted using a haemocytometer. 20,000 single cells were loaded onto each channel of a Chromium chip before encapsulation on the Chromium Controller (10x Genomics, USA). The single-cell sequencing libraries were generated using the Single Cell 5′ V.1, as per the manufacturer's protocol. Libraries were sequenced using Illumina NovoSeq 6000, using Novaseq Control Software V1.7, to achieve a minimum depth of 50,000 raw reads per cell. The libraries were sequenced using the following parameters: Read1: 26 cycles, i7: 8 cycles, i5: 0 cycles; Read2: 98 cycles to generate 75 bp paired-end reads.

**Single-cell RNA sequencing data generation and annotation**. Sequencing data were demultiplexed and quantified using the Cellranger tool (version 4.0.0, 10× Genomics) and aligned to the combined human (official Cell Ranger reference, GRCh38-2020-A) and SARS-CoV-2 reference transcriptomes (Ensembl reference

Sars_cov_2.ASM985889v3). CellBender (version 0.2.0)[76] was applied to the output from Cell Ranger software after alignment to remove background effect from ambient mRNA released during processing. Doublet detection and exclusion was performed using Scrublet (version 0.2.1) with thresholding of cells with a doublet score above two median absolute deviations from the median. Low-quality cells were removed using thresholds of < 200 genes and > 20% mitochondrial content. The analysis was performed using Seurat (version 4.0.1). Data were normalised and log-transformed using NormalizeData and the top 2000 variable genes identified using the FindVariableFeatures tool. The first 20 principal components were batch-adjusted using Harmony (by sample ID) and used to generate the nearest-neighbour graph. Dimensionality reduction and embedding was performed using Uniform Manifold Approximation and Projection (UMAP), with the neighbour-hood graph clustered using the Leiden algorithm. The Wilcoxon rank sum test ($\log_2$ fold change threshold of 0.25, adjusted $P$ value of 0.05) was used to identify differentially expressed genes between clusters, and these were annotated based on expression of markers from literature. This annotation was validated by comparing to a recently published scRNA-seq dataset from ex vivo primary nasal cells[15]. The Seurat label transfer tool was used to assign predicted identities to our data using the external published data as reference. The robustness of our annotation was then assessed by the strength of its correlation with this prediction.

**Gene set scoring and gene set enrichment analysis**. A published gene set derived from IFN alpha or IFN gamma-treated human nasal basal cells[11] was used to generate a list of epithelial-specific IFN-stimulated genes (ISGs). To calculate basal expression of ISGs within the unexposed cells, over-expression of this gene list was assessed using the Seurat AddModuleScore tool. Differences between clusters was compared by a two-tailed Wilcoxon rank sum test (adjusted alpha 0.05) with Benjamini-Hochburg multiple test correction applied. Gene set enrichment analysis was performed using the fgsea tool[77]. Genes were ordered between mock-infected and infected cells by fold-change in expression with Wilcoxon rank sum testing using the FindAllMarkers function in Seurat, but without thresholds. Gene sets from Hallmark, Reactome and Biocarta were used as reference after filtering to exclude those with fewer than 50 and greater than 200 genes. Output from fgsea was further filtered to remove pathways that were not significantly enriched in any cell type (adjusted $P$ value < 0.05) followed by further manual curation of the resultant pathways.

**Regulon scoring and analysis**. The DoRoThEa/Viper package in R studio (Version 3.6.2) was used to score regulon activity by cell[78]. Human regulons from DoRoThEa were filtered for those with a confidence score A-C. Normalised enrichment scores (NES) for each transcription factor (TF) were calculated using run_viper with a minimum regulon size of 4 on the complete gene expression matrix. To estimate TF activity over baseline within each cell type in the infected cells, the median NES from the mock-infected clusters was subtracted from the equivalent in the infected cluster. These scores were then scaled within each TF to give comparative estimate of TF activity between clusters.

**Quantitative RT-PCR**. Total RNA was isolated using TRIzol reagent (Invitrogen, Carlsbad, CA, USA) according to the manufacturer's instructions. For RT-PCR analyses of transcripts, 250 ng of RNA isolated from the nasal epithelial cells was reverse transcribed with Superscript III (ThermoFisher Scientific), and the resulting cDNA templates were subjected to qPCR with a TaqMan Gene Expression Master Mix (Applied Biosystems, MA, USA) and AriaMx real-time PCR system (Agilent Technologies, CA, USA) using Aligent Aris Mx software (version 1.6) according to the manufacturer's instructions. The following TaqMan gene expression assays (Thermo Fischer) were used: *IFNA1* (Hs03044218_g1), *TNF* (Hs00174128_m1). The primers were designed using the Roche Universal Probe Library (UPL) Assay Design tool (Roche, Basel, Switzerland) with the indicated UPL probes. All other primer and probe information is described in Supplementary Table 5. Cycling conditions were as follows: reverse transcription at 50 °C for 15 min, followed by initial polymerase activation at 95 °C for 10 min, then 40 cycles of denaturation at 95 °C for 15 s and annealing/extension at 60 °C for 1 min The $2^{-\triangle\triangle Ct}$ method was used to calculate the relative expression of genes. Each sample was run in duplicate. Samples were normalised to the endogenous housekeeping gene expression, either *RNASEP* for *N* gene expression, or *18S* for all other genes.

**Immunofluorescence**. Infected and mock-infected membranes were fixed in situ with 4% (w/v) paraformaldehyde overnight at 4 °C, before removal from transwells and sectioning. Membranes were washed twice for 5 min in PBS plus 0.1% (v/v) Triton X-100 (Sigma-Aldrich) before being blocked with a 1% (w/v) BSA solution in PBS with 0.5% (v/v) Tween 20 (Sigma-Aldrich; PBST) for one hour at room temperature (RT). Membranes were incubated with primary antibodies for 2 h at RT. Antibodies used are listed in Supplementary Table 6. Membranes were washed three times for 5 min in PBST before incubation with appropriate fluorescence-conjugated secondary antibodies for 2 h at RT. This process was repeated as part of a sequential staining process where required. Membranes were washed three times for 5 min in PBST before being incubated with DAPI (50 nM; Sigma-Aldrich) as a nuclear counterstain, and phalloidin, DyLight 650 (1 unit/mL; ThermoFisher) where required, for 10 min Membranes were then mounted in MOWIOL

mounting media (Sigma-Aldrich) and coverslips applied. Appropriate secondary only controls were performed as required. Images were captured using a Nikon A1 confocal microscope using NIS-Elements C Software (Nikon, Japan), with all capture settings standardised. For analysis three random fields per sample were captured at ×20 magnification and analysed via ImageJ (Version 2.0) using the Cell Counter plugin. Total cell count, and number of spike protein-positive cells, were assessed by segmenting images using ZO-1 and DAPI to identify individual cells. Mean pixel intensity of the spike protein in positive cells was also assessed using the Plot Profile plugin.

**Immunoblot**. Proteins from cell lysates were separated by 10% sodium dodecyl sulphate polyacrylamide gel electrophoresis (SDS-PAGE) using MOPS running buffer (Thermo Fisher, USA), and transferred to polyvinylidene difluoride (PVDF) membranes (Millipore, USA) using NuPage Tris-Bis Transfer Buffer (Thermo Fisher) for immunoblotting (for details of antibodies see Supplementary Table 6). Blots were developed with Pierce ECL Western blotting substrate (Thermo Fisher) and imaged on a LI-COR Odyssey Fc (LI-COR, USA) using Image Studio software (version 5.2.5, LI-COR). Densitometry was undertaken also using Image Studio software (version 5.2.5, LI-COR).

**Transmission electron microscopy**. Cultures were fixed with 2% glutaraldehyde (Sigma-Aldrich, MO, USA) in 0.1 M sodium cacodylate (pH 7.4) buffer in the apical and basal compartment and then kept at 4 °C overnight. For TEM resin processing, the monolayer membranes were removed from the insert frame and placed in microwave sample holders. The Pelco Biowave Pro+ microwave (Pelco, CA, USA), incorporating the Pelco ColdSpot Pro system, was used for the following steps of the processing. The ColdSpot system improves inconsistent wattage supply to the microwave compartment, therefore protecting samples from excess microwave energy. The range temperature was set at 23–27 °C. Following buffer rinses (three pulses at 150 watts (W) for 40 s) the samples were post-fixed in 1% osmium tetroxide for 8 min [pulse microwaved (MW), 100 W] and rinsed in distilled $H_2O$ (three times at 150 W for 40 s, per step). Samples were dehydrated in a graded series of acetone (25%; 50%; 75%; three times with 100% (v/v); 150 W, 40 s per step) before being impregnated with increasing concentrations of epoxy resin (medium resin; TAAB, UK) in acetone (25%; 50%; 75%; three times at 100% (v/v); 300 W, 3 min per step). The samples were then embedded in 100% fresh resin and left to polymerise at 60 °C in a conventional oven for a minimum of 24 h. All resin blocks were trimmed using a razor blade to form a trapezoid block face. Sections were cut on an ultramicrotome using a diamond knife. Semi thin sections (0.5 μm) were stained with toluidine blue and viewed on a light microscope to verify presence of cell monolayers. Ultrathin sections (70 nm) were then cut and picked up onto pioloform-coated copper grids. Grids were stained with 1% (w/v) uranyl acetate (30 min) and 3% (w/v) lead citrate (7 min) to improve contrast. All sections were examined using a HT7800 120 kV TEM (Hitachi, Japan). Digital micrographs were captured using an EMSIS Xarosa CMOS Camera with Radius software (version 2.1, EMSIS, Germany). ImageJ (FIJI) software (Version 5.2.5) was used to enhance the contrast by increasing the percentage of saturated pixels to 1% to aid virus-like particle identification. Virus-like particles (~70 nm in diameter) were counted using the cell counter plugin in at least 6 goblet cells and 16 ciliated cells per donor. Data were presented as number of virus-like particles per cell.

**Proteome sample preparation**. The protein concentration was measured by EZQ protein quantification assay. Protein digestion was performed using the S-Trap sample preparation method and TMT-16 plex labelling was carried out as per the manufacturer's instructions. Samples were cleaned using MacroSpin columns, and dried down prior to offline high-performance liquid chromatography fractionation. Peptides were fractionated on a Basic Reverse Phase column on a Dionex Ultimate 3000 off-line LC system. A total of 18 fractions were collected, and each fraction was acidified and dried. Peptides were dissolved in 5% formic acid, and each sample was independently analysed on an Orbitrap Fusion Lumos Tribrid mass spectrometer, connected to an UltiMate 3000 RSLCnano System. Data collection was done using UltiMate 3000 RSLCnano system software. All spectra were analysed using MaxQuant v1.6.10.43 and searched against SwissProt *Homo sapiens* and Trembl SARS-CoV-2 FASTA files. Reporter ion MS3 was used for quantification and the additional parameter of quantitation labels with 16 plex TMT on N-terminus or lysine was included. A protein and peptide false discovery rate (FDR) of less than 1% was employed in MaxQuant. Moderated t-tests, with patient accounted for in the linear model, was performed using Limma, where proteins with an adjusted P < 0.05 were considered as statistically significant. All analysis was performed using R studio (Version 3.6.2). Raw data, including exact P values, are present in Supplementary Data 5. A comprehensive description of the methods can be found in the Supplementary Methods.

**Statistical analysis**. Statistical analysis was performed and figures assembled using GraphPad Prism V9 (GraphPad Software, USA). Data are presented as mean ± SEM of individual donor values (derived typically from 2 to 3 independent repeat experiments per donor). The donor was used as the unit of experiment for statistical analysis purposes. Continuous data were normalised or log-transformed prior to analysis using parametric significance tests, or if this was not possible, were

analysed using nonparametric significance tests. Differences between two groups were compared using an unpaired, two-tailed Student's *t*-test (or Mann–Whitney test for TEM image analysis), whereas differences between more than two groups used ANOVA, with Dunnett's post-test correction for multiple comparisons when comparing to a single reference point (e.g. mock-infected or time zero) or with Sidak's post-test correction for other multiple comparisons (e.g. between differently treated donors at the same time point). In some cases, data were normalised to a reference point, where a one-sample t-test was used. Unless stated otherwise, all tests were two sided and an alpha of < 0.05 was the threshold for statistical significance. Statistical analysis of proteomics and transcriptomics datasets is described in the relevant sections above.

**Reporting summary**. Further information on research design is available in the Nature Research Reporting Summary linked to this article.

## Data availability

The mass spectrometry proteomics data generated in this study have been deposited to the ProteomeXchange Consortium via the PRIDE partner repository with the dataset identifier PXD022523. The proteomic analysis in this study involved using reference proteomes from the following databases: SwissProt Homo sapiens under the proteome ID: UP000005640 9606 and Trembl SARS-CoV-2 FASTA files under the proteome ID: UP000464024. Raw RNA sequencing data have been deposited to the European Genome-Phenome Archive (study ID:) EGAS00001005633. Processed scRNAseq data is available at Zenodo (https://zenodo.org/record/4564332, https://doi.org/10.5281/zenodo.4564332). RNA Sequencing data was aligned to the combined human transcriptome using the official Cell Ranger reference under the assession code GRCh38-2020-A and the SARS-CoV-2 reference transcriptomes are available in Ensembl under the reference code Sars_cov_2.ASM985889v3 (https://covid-19.ensembl.org/index.html). The clustering annotation was validated using Seurat label transfer from a published scRNA-seq dataset from nasopharyngeal swabs (https://doi.org/10.1016/j.cell.2021.07.023). ISG signature scores were generated using context-specific ISGs from a published IFN-treated nasal cell dataset (https://doi.org/10.1016/j.cell.2020.04.035). Gene sets from Hallmark (gsea-msigdb.org), Reactome (reactome.org) and Biocarta (biocarta.com) datasets were used as reference after filtering to exclude those with fewer than 50 and greater than 200 genes. List of figures with associated raw data: Fig. 1a–h, Fig. 2d, e, Fig. 3a–c Fig. 4a–f, Fig. 5a–c, Fig. 6a–e, Fig. 7a–e, and supplementary Fig. 2, supplementary Fig. 7, supplementary Fig. 8, supplementary Fig. 10. Source data includes uncropped blots, all quantitative data and PFU counts. The results of differential expression analysis of RNA-seq and proteomics data are included as supplementary datasets. Additional raw data are available on request from the corresponding author providing ethical approvals permit sharing of data. Source data are provided with this paper.

## Code availability

Analysis scripts and code[79] are available at github.com/haniffalab/covid_nasal_epithelium.

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

## Acknowledgements

We acknowledge Public Health England for providing the SARS-CoV-2 isolate and Professor R. Randall (St Andrew's University) for providing Sendai and parainfluenza 3 viruses and the vero E6 cell line. We thank M. Glanville and Dr. S. Dainty (Newcastle University Infectious Disease Facility) and Dr. A. Laude (Bioimaging facility) for assistance. Tissue for this study was provided by the Newcastle Biobank which is supported by the Newcastle upon Tyne NHS Foundation Trust and Newcastle University. The work was part-funded by the Barbour Foundation (C.J.A.D., S.H., M.B., M.T., M.H., C.W.), the UK-Coronavirus Immunology Consortium (C.J.A.D., S.H., M.H.) and the Medical Research Council SHIELD antimicrobial resistance consortium (J.S., A.J.S.). C.F.H. is supported by a Medical Research Council studentship (MR/NO13840/1) and M.B. by an MRC Clinician Scientist Fellowship (MR/M008797/1). M.T. and S.H. are funded by Wellcome Investigator Awards (215542/Z/19/Z and 207556/Z/17/Z). C.J.A.D. and G.R. are Wellcome Clinical Research Career Development Fellows (211153/Z/18/Z and 214539/Z/18/Z). M.E.D. is a Marie Sklodowska Curie Fellow within the European Union's Horizon 2020 research and innovation programme under the Marie Sklodowska-Curie grant agreement No. 890296. M.H. is funded by Wellcome (WT107931/Z/15/Z), The Lister Institute for Preventive Medicine and Newcastle NIHR Biomedical Research Centre (BRC). J.P.G., C.W. and B.V. were supported by the Medical Research Foundation (MRF Respiratory Diseases Research Award to J.P.G.; Grant MRF-091-0001-RGGARNE) and Boehringer Ingelheim. TEM work was supported by a BBSRC Alert17 grant (BB/R013942/1). F.G. was supported by the Bubble Foundation. A.J.S. is a National Institute for Health Research (NIHR) Senior Investigator. The views expressed in this article are those of the author(s) and not necessarily those of the NIHR, or the Department of Health and Social Care. The funders had no role in the study design, data collection and analysis, decision to publish, or preparation of the manuscript.

## Author contributions

Conceived the study: M.B., C.W. and C.J.A.D. with M.H., S.H. and M.T. Experimental design: M.H., S.H., M.B., C.W., M.T., G.R. and C.J.A.D. Nasal model development and patient material: I.J.H., B.V., J.S., J.P.G., S.C., J.P., A.J.S., M.B. and C.W. Virology data generation, analysis and interpretation: C.F.H., B.J.T., J.S.S., F.G., A.I.G., C.M.A.K., L.H., T.D., S.H. and C.J.A.D. Proteomics data generation, analysis and interpretation: M.E.D., S.M. and M.T. Single-cell sequencing data generation, analysis and interpretation: R.A.B., E.S., R.H., J.C., M.H. and G.R. Supervised research: A.J.S., M.H., S.H., M.B., C.W., M.T., G.R. and C.J.A.D. Drafted the manuscript: C.F.H. and C.J.A.D. with E.S., B.V., M.E.D., T.D., M.T. and G.R. Revised the manuscript: R.A.B., M.E.D., I.J.H., J.S.S., A.J.S., M.H., S.H. and C.W. Approved the manuscript for submission: all authors.

## Competing interests

S.H. declares honoraria from CSL Behring and Takeda for teaching and consultancy. M.B. declares being CI on investigator-led research grants from Pfizer and Roche Diagnostics; speaker fees paid to Newcastle University from Novartis, Roche Diagnostics and TEVA; travel expenses to educational meetings from Boehringer Ingelheim and Vertex. The remaining authors declare no competing interests.
