## [Peer Review File · Nature Communications]

Delayed induction of type I and III interferons mediates nasal epithelial cell permissiveness to SARS-CoV-2REVIEWER COMMENTS

Reviewer #1 (Remarks to the Author):

This review was prepared and is signed by Dr. Jose Ordovas-Montanes.

Summary Statement

In the study by Hatton et al., the authors set out to address the question of how the initial host interactions between SARS-CoV-2 may play out in human nasal epithelial cells. This is an important question as most current studies have relied on cell lines, or pseudotyped viruses, and have not utilized clinical isolates in conjunction with primary epithelial cells. The study utilizes scRNA-seq and proteomics of primary nasal epithelial cultures in ALI. The authors found that SARS-CoV-2 has widespread tropism for various nasal epithelial cell types, that Type I and III IFNs and ISGs are induced, and that there is a relative delay with respect to viral gene expression. The authors conclude that recombinant IFNs may be useful to induce an efficient antiviral state. This study is timely, important and interesting as it extends our knowledge beyond in vitro models and brings us closer to what may be occurring in vivo in the human nasal epithelium. If the authors are able to address the two major comments and provide further clarity and context as mentioned in the minor comments, the study is definitely suitable for publication. The study is generally well communicated and accurately conveyed save for a few claims of “first” which due to the rapidly evolving field we may suggest the authors reduce.

Major comments

o The reliable classification of infected and bystander cells is key. It is important to tune the parameters of what constitutes an infected cell relative to the background identified in a sample. We suggest the use of CellBender to clean up the SARS-CoV-2 reads specifically, as if not the background can be difficult to interpret. A re-analysis of figures dependent on this classification should be performed (<https://github.com/broadinstitute/CellBender>). Alternatively, different UMI cutoffs should be used to bin cells into viral infection categories (high/mid/low) and identify if differential expression and results reported for cell type infection bias are robust.

o The spike protein imaging as in Figure 2D could be better explained and/or controlled. The anti-spike antibody appears to maybe have some background. Could the authors identify cells that were determined to have positive or negative staining? Is a control uninfected ALI included for reference and correction of background pixel intensities vs. real staining?

Minor Comments

o For reproducibility, it would be helpful to include full description of how the nasal airway epithelial cells are processed, differentiated and validated, including the specific modifications mentioned here in the methods rather than referring to prior work.

o Author's should provide a table for differential expression comparisons to accompany scRNA-seq data represented as dot plots as if not significance is hard to conclude for these plots.

o If there is information on the key mutations in England/2/2020 that have been focused on in the literature, primarily in the context of immune escape, it would be beneficial to include this info to provide context for this study moving forward. i.e. what is the publicly-appreciated strain name or mutations this virus contains?

o The authors should specify biosafety levels and precautions used for working with clinical isolates of SARS-CoV-2.

o Catalog number should be included for cytokines

o We appreciate the detailed methods section, especially the scRNA-seq sample processing. This may be a question for the editors more than the authors, but is there any way of having all of the methods together rather than separated out? It would make the methodology much easier to find and follow.

o What threshold was used for the Wilcoxon rank sum test? For log-fold change and for statistical value interpreted as significant?

o Figure legends should state cell numbers, precise timing, and total sample numbers included in each experiment for clarity. There are several figure legends where this is missing or unclear: What was the timepoint for Figure 2A? How many samples were assessed in 2B?

o Figure 3C is somewhat unclear as it mentions differentially expressed IFNs by cell type, but many of the genes appear to be almost not detected, so only some of these are differentially expressed?

o Could the authors please clarify why ACE2, no TMPRSS2 and ACE2, no Furin have distinct expression frequencies in Supp Figure 1? Shouldn't these both be the same as they are just representing ACE2? Is the joint expression an "or" gate or an "and" gate in terms of frequency of cells reported? The figure legend could better explain the methodology used.

o The Foxn4 cells are likely the recently-described deuterosomal cell type. Would suggest the authors rename accordingly and cite Garcia et al., Novel dynamics of human mucociliary differentiation revealed by single-cell RNA sequencing.... In Development <https://dev.biologists.org/content/146/20/dev177428>

o Would suggest the authors contextualize the study within recent pre-prints that have come out in the discussion. This by no means impacts novelty of the work here but rather is helpful for the community to place the author's work in the rapidly evolving context. Would suggest to discuss in light of the in vivo and in vitro work in Cheemarla et al., Magnitude and timing of the antiviral response determine.... medRxiv

<https://www.medrxiv.org/content/10.1101/2021.01.22.21249812v2.full-text> and Ziegler et al., Impaired local intrinsic immunity to SARS-COV-2 infection in severe COVID-19... bioRxiv

<https://www.biorxiv.org/content/10.1101/2021.02.20.431155v1.full> and Ravindra et al., Single-cell longitudinal analysis of SARS-CoV-2 infection in human airway epithelium identifies target cells... PLOS Biology <https://pubmed.ncbi.nlm.nih.gov/33730024/> and Purkayastha et al., Direct exposure

to SARS-CoV-2 and cigarette smoke increases infection severity and alters.... Cell Stem Cell
<https://www.ncbi.nlm.nih.gov/pmc/articles/PMC7402049/>

o Would suggest the authors provide a reference for lines 182-184 on ISG effective by paracrine and the mechanisms proposed.

o Line 203-204: IFN genes are very transiently expressed and difficult to detect by RNA-seq, and especially by scRNA-seq. This small minority may not accurately reflect IFN gene induction. Suggest that authors discuss some of these caveats.

o More details should be provided on Figure 5c regarding generation and interpretation of the proteomics based network.

o Failure to exert control may be too large of a claim by 72 hours. The kinetics of control may be delayed and could be investigated. Would suggest the authors either tone down claims or provide experimental data beyond 72 hours.

Reviewer #2 (Remarks to the Author):

In this manuscript, Hatton et al. studied SARS-CoV-2 infection and the subsequent interferon response of nasal epithelium using an organotypic air-liquid interface model. The authors applied single-cell RNA sequencing to determine the infection of different epithelium cell subsets. In addition, they determined the production of type I and III IFNs, and the subsequent ISG response over time. Finally, they showed that recombinant IFN- β or IFN- λ 1 restricts viral replication.

The manuscript is written in a clear and understandable manner. Nasal epithelium is very relevant tissue to study in the context of SARS-CoV-2, since it is probably the main route of initial infection. The used organotypic model is elegant, and closely mimics *in vivo* differentiated cells. The findings that SARS-CoV-2 can infect nasal epithelium, and that this can be partially counteracted by prior treatment with recombinant type I and III IFNs, are very much in line with previous findings and general expectations. The main novelty of this manuscript (as also stated in the title) would be the finding of a delayed type I and III IFN response by nasal epithelium upon SARS-CoV-2 infection. Yet, this reviewer has doubts whether the data support this conclusion of the authors, for the following reasons:

- First, for many viruses, substantial levels of infection are required before intracellular microbial structures rise above the threshold of detection in epithelial cells. As shown in Fig. 1E, it takes 24h to reach substantial levels, which peak at 48h. In that regard, it is questionable whether the IFN response by these cells is really delayed.

- Second, and more important, it is completely unclear whether this (supposed) delayed IFN response is specific for SARS-CoV-2. The response to SARS-CoV-2 was only compared to one other

virus, Sendai virus, which is not only very different from coronaviruses, but also very different regarding pathogenicity. Before the authors can conclude that SARS-CoV-2 shows a delayed IFN response by nasal epithelium, it is essential that they compare this response to IFN production by other (related) viruses.

In addition, it would be nice if the authors could try to put the findings of this study into perspective, particularly as to whether it would be related to disease severity. Patients only progress to severe disease at about two weeks post infection. Would that still be related to differences in infection and/or IFN responses in nasal epithelium? Are differences in infection and/or IFN responses to be expected between individuals, e.g. pediatric vs adult, related to obesity, or other risk factors?

Reviewer #3 (Remarks to the Author):

In this work, Hatton and co-workers (NCOMMS-21-06326) employ scRNAseq on SARS-CoV-2 infected ALI cultures to address the antiviral response which is generated by primary airway epithelial cells. They show that most cell types are susceptible to SARS-CoV-2 infection and that at 24hpi, infected cells produce ISGs while the neighbours non-infected cells (bystander cells) do not. The authors suggest that this limitation of ISG production to infected cells might be the consequence of both a limited production of IFN which appear limited to a small fraction of ciliated cells and a delayed production of IFN (scRNAseq performed at 24hpi). They suggest that the ISG produce in the infected cells are direct ISG, made directly downstream IRF3 without the need of IFN receptor-mediated signalling. Using mass spectrometry approaches, the authors could show that SARS-CoV-2 infected ALI cultures mount a “typical” IFN mediated response (production of classical ISG) at 72 hpi. Finally, the authors show that treatment of cells with RUX to block the IFN-mediated response did not render cells more permissive to SARS-CoV-2 infection while pre-treatment of ALI cultures with IFN partially protected against infection. Together the authors suggest that the delayed IFN response of ALI cultures to SARS-CoV-2 infection is not sufficient to protect the cells against SARS-CoV-2.

While I appreciate the novelty in using scRNAseq on infected ALI cultures, I fail to identify what are the new findings in this work. The fact that most cell types could be infected by SARS-CoV-2 has been previously described and the authors refer to some of these papers and similarly, the fact that the response to viral infection leads to a delay production of IFN which fail to control SARS-CoV-2 was already reported in primary airway epithelial cells culture (Rebendenne et al. 2021). The authors observe that at 24hpi, infected cells produce ISGs while they failed to detect IFN (or very little). The authors speculate that these ISGs are direct ISG made directly downstream IRF3. This is potentially really interesting, but the authors should demonstrate this hypothesis and identify how can at early time post infection, ISG are made directly downstream IRF3 while at later time post infection (48hpi Figure 6), ISG will be made downstream the IFN receptors.

My detailed comments to this work are:

Major comments

- Using single cell RNAseq, the authors show that they were able to recover the major cell types of upper respiratory tract (Ciliated, Secretory, Basal etc) based on their gene expression profiles and immunostaining of selected marker genes. Line 108: what do the authors mean by corresponded broadly to ex-vivo scRNA seq data. It is important that the authors perform a qualitative and quantitative evaluation of the gene expression profiles of their ALI culture compared to biopsies, to judge how similar their ALI models are compared to primary airway epithelial cells in-vivo. A systematic comparison with cell types identified from previously published single cell RNAseq studies of primary upper respiratory tract tissues is missing. Such an analysis can be easily implemented with label transfer methods like those implemented in Seurat to exhaustively annotate cell types based on prior knowledge from related primary tissues.
- In Line 150 - the authors write - "A greater proportion of ciliated and secretory cells expressed viral transcripts compared to basal cells", however looking at Fig 2(A,B), the cell subtype FOXN4 also seems to express the viral transcripts at similar levels, in-fact higher than secretory cells and at similar levels to ciliated. Surprisingly, this rare cell type is hardly discussed in the manuscript. The authors should explain the reasons behind this. A complementary study from Ziegler et. al, <https://doi.org/10.1101/2021.02.20.431155> (Bioarxiv preprint), has shown that the FOXN4+ cells are affected upon SARS-Cov2 infection, thus in my opinion, this cell type warrants further investigation in the ALI infection model system.
- INF gene sets - The authors have used a generic ISG gene list from MsigDB to score the interferon response in unexposed, uninfected and infected cells, however, interferon-stimulated gene lists specifically from primary nasal epithelium tissues and ALI models treated with different interferons in dose dependent manner exists. Using such context specific ISGs is probably better to evaluate the immune response generated by the ALI culture
- Instead of simply looking at the expression of regulators of INF response, it might be insightful to perform gene regulatory analysis using methods like SCENIC to identify the transcription factor activity of relevant TF's like IRF's, JAK/STATS in the infected cells and corroborate some of those with the proteomics results.
- It is very difficult to appreciate the immune response generated by the different cell types from the heatmap shown in figure 3B. I am expecting the data to be the relative expression compared to mock infected cells. I could not find information about the expression value, these are Log values? Figure 3B should be complemented with a supplementary figure that plot all ISG expression level for a cell type (with the X-axis being mock, bystander and infected). This will allow the reader to have a better view of ISG expression and whether they are up or downregulated upon infection. From the scRNAseq data, it looks like the level of some ISG expression is even down regulated in the bystander cells. The authors should explain this rather unusual inhibition of ISG production in bystander cells?

Can SARS-CoV-2 inhibit IFN-mediated response in bystander cells by secreting viral/cellular factors?
This should be address experimentally.

- the authors suggest that the fact that only infected cells display ISG at 24hpi in their scRNAseq data is the consequence of direct ISG production (downstream IRF3). According to the data shown in figure 6 using higher MOI and different time post-infection, the lack of ISG in bystander cells may be due be the result of the time point used for the scRNAseq (24hpi), too soon to detect paracrine signalling. The authors should rule out this possibility. For example, the experiment shown in figure 6 with RUX should be repeated at earlier time post-infection to show that the production of ISG in the infected cells is not due to IFN-mediated signalling but to a direct production of ISG via IRF3.

- Why do the authors only focus on type I IFN for the exogenous IFN treatment experiment at different time pre- and post-infection? The authors should also add exogenously type III IFN to address which IFN (Type I or type III OFN) would be best as therapeutic approaches

Minor comments:

- line 125: the decline of TEER can also be due to cell death, the authors should add a little statement about this in their text. Could the authors provide the non-cut version of their WB in supplementary? This will be very useful for reference purposes for other lab to appreciate the background of the antibodies used the authors should label their plots with the name of the different cell types but not the cell type specific markers as it will make it easier for the reader to quickly know what cell type the plot is about

- line 302, on the WB figure 6A, there is a greater band for the lower one in the presence of RUX. Is this non-glycosylated spike? Because the amount increase with RUX.

- Data availability - Given that the data generated in this study will be very useful to the larger community, in addition to submitting the raw data to standard repositories, the authors should also consider making their processed data from their single cell RNAseq analysis (counts and cell annotation tables as Seurat objects) and Proteomics analysis (Normalized log2 transformed tables as CSV table) easily available. These non-restricted data can be provided as supplementary material or through third party hosting options like Figshare or Github.

- Data Reproducibility - In interests of data reproducibility, transparency and open science, the authors should also share their scripts for performing the computational analysis related to single cell RNAseq and proteomics experiments shown in this manuscript in a code sharing platform like GitHub/GitLab etc

- Standard QC metrics from Cell ranger (number of reads, reads per cell, alignment rate, cell counts etc) and QC plots from Seurat (nCount, nUMI, %MT etc) pertaining to the single cell RNAseq samples should also be provided as supplementary tables.

Reviewer 1

Summary Statement

In the study by Hatton et al., the authors set out to address the question of how the initial host interactions between SARS-CoV-2 may play out in human nasal epithelial cells. This is an important question as most current studies have relied on cell lines, or pseudotyped viruses, and have not utilized clinical isolates in conjunction with primary epithelial cells. The study utilizes scRNA-seq and proteomics of primary nasal epithelial cultures in ALI. The authors found that SARS-CoV-2 has widespread tropism for various nasal epithelial cell types, that Type I and III IFNs and ISGs are induced, and that there is a relative delay with respect to viral gene expression. The authors conclude that recombinant IFNs may be useful to induce an efficient antiviral state. This study is timely, important and interesting as it extends our knowledge beyond in vitro models and brings us closer to what may be occurring in vivo in the human nasal epithelium.

We thank the reviewer for this supportive statement on the significance of our findings.

If the authors are able to address the two major comments and provide further clarity and context as mentioned in the minor comments, the study is definitely suitable for publication. The study is generally well communicated and accurately conveyed save for a few claims of “first” which due to the rapidly evolving field we may suggest the authors reduce.

We are grateful to the reviewer for their comments, all of which we have addressed and which we believe have improved the manuscript. These changes are summarised below.

Major comments:

1. The reliable classification of infected and bystander cells is key. It is important to tune the parameters of what constitutes an infected cell relative to the background identified in a sample. We suggest the use of CellBender to clean up the SARS-CoV-2 reads specifically, as if not the background can be difficult to interpret. A re-analysis of figures dependent on this classification should be performed (<https://github.com/broadinstitute/CellBender>). Alternatively, different UMI cutoffs should be used to bin cells into viral infection categories (high/mid/low) and identify if differential expression and results reported for cell type infection bias are robust.

We thank the reviewer for this suggestion. We have now used CellBender to clean up reads and performed all downstream analysis, and re-analysed figures, using the cleaned data (Fig. 1A-C, Fig. 2A-B, and Fig 3A-C). The findings remained fundamentally unchanged but we find, having removed signal from ambient RNA, better resolution between infected and uninfected cells. Importantly this also better allows us to determine the relative levels of infection within each cell type.

For ease of reference, the relevant figure panels are included below:

Fig. 1A-C

Fig 2A-B

Fig. 3A-C

Discussion of the revised analysis of Fig. 3 is included in responses to the individual comments below (see Reviewer 1, minor comment 9; Reviewer 3, major comments 3-5).

2. The spike protein imaging as in Figure 2D could be better explained and/or controlled. The anti-spike antibody appears to maybe have some background. Could the authors identify cells that were determined to have positive or negative staining? Is a control uninfected ALI included for reference and correction of background pixel intensities vs. real staining?

We performed parallel staining of uninfected ALI cultures in this analysis, which revealed no evidence of background staining. On this basis, pixel intensities were not corrected for background. These data are included in a new Supplementary Figure S4:

Minor comments:

1. For reproducibility, it would be helpful to include a full description of how the nasal airway epithelial cells are processed, differentiated and validated, including the specific modifications mentioned here in the methods rather than referring to prior work.

We have included this more detailed description in the methods as requested:

*“Tissue shaved from the superficial surface of the sample was chopped into ~2 mm² pieces and added to RPMI-1640 basal medium containing 0.1% protease (Sigma-Aldrich, UK) and incubated overnight with gentle agitation at 4°C. All large pieces of tissue were discarded, and residual protease was neutralized with 5% FCS. The preparation was centrifuged (200 g; 7 min) and the pellet resuspended in PneumaCult-Ex Plus expansion medium (Stemcell Technologies), then seeded onto 25 cm² tissue culture flasks pre-coated with 30 µg/mL Type I collagen (PureCol, Advanced BioMatrix). Flasks were incubated in a humidified atmosphere containing 5% CO₂ at 37°C, with medium replaced every 48 hours. Cells were trypsinised at 60-80% confluence and cryopreserved for future use. Upon thawing, cells were grown through an additional expansion phase, then transferred in Ex Plus medium onto collagen-coated 6.5 mm polyester transwell membranes with 0.4 µm pore size (Corning) at a density of 150,000 cells/cm². When cells were fully confluent, apical medium was removed and basolateral medium was switched to PneumaCult-ALI-S (Stemcell Technologies). Cells were maintained at air-liquid interface until fully differentiated. Barrier integrity of ALI cultures was monitored by measuring trans-epithelial electrical resistance (TEER, EVOM 2, World Precision Instruments). ALI cultures were validated for use in experiments based on microscopic appearance of appropriate ciliated morphology and TEER > 500 Ω*cm². The sex and age of donors are included in Table S4.”*

2. Author’s should provide a table for differential expression comparisons to accompany scRNA-seq data represented as dot plots as if not significance is hard to conclude for these plots.

A full list of DEGs is now included in supplementary datasets as follows:

- i) DEGs between all cell types (dataset S1), between all viral exposed cells (dataset S2),
- ii) DEGs between uninfected and bystander or infected cells within clusters (datasets S3 and S4).

3. If there is information on the key mutations in England/2/2020 that have been focused on in the literature, primarily in the context of immune escape, it would be beneficial to include this info to provide context for this study moving forward. i.e. what is the publicly-appreciated strain name or mutations this virus contains?

The SARS-CoV-2 England/2/2020 is a clinical isolate from an individual infected in the UK in January 2020. There is limited published information on this strain (see Conceicao et al PLOS Biol 2020 <https://doi.org/10.1371/journal.pbio.3001016> and Burton et al, J Virol Methods 2021 DOI: 10.1016/j.jviromet.2021.114087). It is not known to harbour mutations associated with variants of concern. This statement is now included in the methods.

4. The authors should specify biosafety levels and precautions used for working with clinical isolates of SARS-CoV-2.

All infections were performed in the Advisory Committee on Dangerous Pathogens (ACDP) hazard group 3 (HG3) facilities by trained personnel. This is now stated in the methods.

5. Catalogue number should be included for cytokines

The following cytokine catalogue numbers have been added into the methods section; IFN1 1598-IL-025, IFN β 1 Avonex NDC 59627-001-03, Rux S1378.

6. We appreciate the detailed methods section, especially the scRNA-seq sample processing. This may be a question for the editors more than the authors, but is there any way of having all of the methods together rather than separated out? It would make the methodology much easier to find and follow.

We have combined these in the main methods section, as suggested.

7. What threshold was used for the Wilcoxon rank sum test? For log-fold change and for statistical value interpreted as significant

Differential gene expression analysis was performed with a Wilcoxon rank sum test (FindAllMarkers or FindMarkers in Seurat) with a log₂ fold change threshold of 0.25 and adjusted P value threshold of 0.05 (figure 1C, 2A and 3B). For GSEA analysis (Fig 3C) Wilcoxon rank sum testing was performed without thresholding on a minimal fold change or expression level. Wilcoxon rank sum testing was also used to compare interferon gene module scores (Fig 3A) between cell types with a Benjamini-Hochburg multiple comparison correction (adjusted P value threshold of 0.05). This is now clarified in the methods.

8. Figure legends should state cell numbers, precise timing, and total sample numbers included in each experiment for clarity. There are several figure legends where this is missing or unclear: What was the timepoint for Figure 2A? How many samples were assessed in 2B?

This information has been included in all figure legends.

9. Figure 3C is somewhat unclear as it mentions differentially expressed IFNs by cell type, but many of the genes appear to be almost not detected, so only some of these are differentially expressed?

Apologies for the lack of clarity. As previously stated (in response to comment 1), through reanalysis using CellBender in data processing and at the request of Reviewer 3 we have revised Fig. 3C. IFNs are not significantly differentially expressed - these data are mentioned in the text and are now included in revised Fig. S6. We have also included the full DEG analysis results as supplementary datasets (as mentioned above).

Revised Fig. S6

Text description:

“Transcripts for IFN-I (IFNB, IFNK, IFNA5) and IFN-III (IFNL1) were not significantly differentially expressed and were detectable in only a small minority (~ 0.4%) of infected secretory cells (Fig. S6).”

10. Could the authors please clarify why ACE2, no TMPRSS2 and ACE2, no Furin have distinct expression frequencies in Supp Figure 1? Shouldn't these both be the same as they are just representing ACE2? Is the joint expression an “or” gate or an “and” gate in terms of frequency of cells reported? The figure legend could better explain the methodology used.

We have revised this figure for clarity to demonstrate the ‘or’ gates and include an explanation of the methodology used (revised Figure S3). The bar charts demonstrate the proportions within each cluster expressing ACE2 and/or the putative coreceptor. Colour indicates each group: neither ACE2 nor coreceptor (green), ACE2 without coreceptor (light blue), coreceptor without ACE2 (purple) and co-expression of ACE2 and coreceptor (dark blue).

Revised Fig. S3.

11. The Foxn4 cells are likely the recently-described deuterosomal cell type. Would suggest the authors rename accordingly and cite Garcia et al., Novel dynamics of human mucociliary differentiation revealed by single-cell RNA sequencing.... In Development <https://dev.biologists.org/content/146/20/dev177428>

We thank the reviewer for this suggestion and have updated the annotation (Fig. 1A-C, Fig. 2A-B, Fig. 3A-C, Figs S3, S5, S6) and references accordingly.

12. Would suggest the authors contextualize the study within recent pre-prints that have come out in the discussion. This by no means impacts novelty of the work here but rather is helpful for the community to place the author’s work in the rapidly evolving context. Would suggest to discuss in light of the in vivo and in vitro work in Cheemarla et al., Magnitude and timing of the antiviral response determine.... medRxiv <https://www.medrxiv.org/content/10.1101/2021.01.22.21249812v2.full-text> and Ziegler et al., Impaired local intrinsic immunity to SARS-CoV-2 infection in severe COVID-19... bioRxiv <https://www.biorxiv.org/content/10.1101/2021.02.20.431155v1.full> and Ravindra et al., Single-cell longitudinal analysis of SARS-CoV-2 infection in human airway epithelium identifies target cells... PLOS Biology <https://pubmed.ncbi.nlm.nih.gov/33730024/> and Purkayastha et al., Direct exposure to SARS-CoV-2 and cigarette smoke increases infection severity and alters.... Cell Stem Cell

We agree and have extensively revised the discussion to incorporate citations to these and other recent relevant studies.

“Our findings indicate that the host response to SARS-CoV-2 in nasal epithelium is dominated by paracrine IFN-I/III signalling, albeit this response is kinetically delayed. These data contrast with initial reports that SARS-CoV-2 did not induce a robust IFN response in airway epithelial cells²⁶⁻²⁸, but are consistent with emerging evidence of IFN-I/III induction in nasal swabs from patients with COVID-19^{15,41-43} and with more recent findings in lung airway models^{35,42,44-46}. Blockade of the endogenous IFN response

had an impact on SARS-CoV-2 infection at later stages post-infection, once the IFN response was established, underscoring the delayed kinetic but also emphasising its functional relevance. While the impact of endogenous IFN-I/III signalling upon SARS-CoV-2 replication has not to our knowledge been investigated previously in nasal cell models, our data are consistent with recent findings in some^{42,46}, but not all epithelial model systems^{27,44}. Our experiments with IAV, PIV3 and SeV - viruses which induced the robust early expression of IFN-I/III, in line with previous studies^{12,27,36} - confirm that this delay was not due to an intrinsic property of nasal epithelial cells.”

Please note this also includes discussion of new data, added in response to the reviewer’s comment (16) below, extending the inhibition of endogenous IFN signalling using ruxolitinib to 96 hpi (see the response to comment 16 below for a fuller description of these data and the results). It also refers to additional experiments performed in response to the request from Reviewer 2.

In addition, we include a new paragraph discussing the wider relevance of the findings in relation to recent ex vivo studies such as those mentioned by the reviewer:

“IFN-I/III signalling is plausibly implicated in protection against life-threatening COVID-19¹⁸⁻²⁰. Consistent with this, circulating immune cells of patients with severe COVID-19 exhibit impaired ISG responses⁵⁵⁻⁵⁷. However, whether the local airway IFN response in the early stages of infection has a decisive role in shaping the subsequent clinical outcome of COVID-19 remains to be conclusively determined. A compelling recent scRNA-seq study reported that patients going on to develop severe disease exhibited a muted ISG response in the nasal airway, in contrast to those with milder disease¹⁵, and is supported by independent findings of attenuated nasal ISG induction in patients with autoantibodies to IFN-I⁵⁸, who are prone to more severe disease²⁰. Additional strands of evidence suggest a potential link between airway IFN-I/III competence and clinical outcome in COVID-19. Age remains the strongest risk factor for poor outcome in COVID-19, and the efficiency of IFN-I/III induction is known to decline with advancing age⁵⁹, and appears to be greater in the nasal airways of children than adults infected with SARS-CoV-2⁶⁰. Other relevant environmental influences, such as exposure to cigarette smoke or other viral infections, are also reported to perturb IFN-I/III responses of airway cells in ways that may be relevant to COVID-19 pathogenesis^{43,61}.”

13. More details should be provided on Figure 5c regarding generation and interpretation of the proteomics based network.

The following text has been added to the Supplemental Materials to address this comment:

‘Proteins with differential abundance (adjusted P value <0.05 and fold change > 1.5) were analysed using the search tool for retrieval of interacting genes (STRING) database version 11 (<https://string-db.org/>). The data was modified for presentation using Cytoscape version 3.7.2. Proteins were grouped by functional categories based Uniprot annotation (www.uniprot.org). In STRING, active interaction sources, including experiments and databases, and an interaction score > 0.7 were applied to construct the protein-protein interaction networks. In the network, the nodes correspond to the proteins identified and the edges represent the interactions. The node colour gradient depicts fold change in protein expression in infected compared to mock samples.’

14. Would suggest the authors provide a reference for lines 182-184 on ISG effective by paracrine and the mechanisms proposed.

In the process of revision this sentence has been removed.

15. Line 203-204: IFN genes are very transiently expressed and difficult to detect by RNA-seq, and especially by scRNA-seq. This small minority may not accurately reflect IFN gene induction. Suggest that authors discuss some of these caveats

We accept these caveats and have acknowledged them in the revised manuscript.

“Transcripts for IFN-I (IFNB, IFNK, IFNA5) and IFN-III (IFNL1) were not significantly differentially expressed and were detectable in only a small minority (~ 0.4%) of infected secretory cells (Fig. S6). Whilst potentially consistent with the absence of paracrine signalling at this timepoint, this might also reflect transient expression and/or insensitivity of detection by scRNA-seq; a similar pattern was observed for other cytokines and chemokines (Fig. S6).”

16. Failure to exert control may be too large of a claim by 72 hours. The kinetics of control may be delayed and could be investigated. Would suggest the authors either tone down claims or provide experimental data beyond 72 hours.

We thank the reviewer for this excellent point. As suggested, we have undertaken additional experiments with ruxolitinib extending our analysis beyond 72 hours. These data show clear evidence of an impact of the endogenous IFN response on viral replication at 96 hpi, using several analogous approaches (Fig. 6B-E). These data are included in a new Figure 6B-E and the text now describes our findings up to 96 hpi.

Figure 6. Impact of endogenous IFN-I/III signalling on SARS-CoV-2 infection. Nasal ALI cultures treated with ruxolitinib (RUX, 10 μ M) or DMSO vehicle for 24 h prior to infection (MOI 0.1). Whole-cell lysates were prepared at the indicated times for RT-PCR analysis of expression of (A) the ISGs USP18, RSAD2 and ISG15 ($n=3$ donors, mean \pm SEM; *** $P < 0.001$, **** $P < 0.0001$, ANOVA with Dunnett's post-test correction compared to mock-infected cells) or (B) viral N mRNA ($n=3$ donors, mean \pm SEM; * $P = 0.035$ ANOVA with Sidak's post-test correction compared to DMSO control). (C) Whole-cell lysates were prepared at 96 hpi for immunoblot analysis of viral S/S2 protein and host RSAD2, USP18 and ISG15 protein expression (representative blot shown of experiments in $n=4$ donors). (D) Densitometry analysis of S/S2 protein intensity relative to GAPDH, normalised to the DMSO control (data from C, $n=4$ donors, mean \pm SEM; ** $P = 0.003$, one-sample t test). (E) Plaque assay of apical washes collected at the times indicated showing a significant increase in infectious particle release at 96 hpi (same experimental

*conditions as C-D; n=4 donors, mean \pm SEM; * $P = 0.015$, ANOVA with Sidak's post-test correction compared to DMSO control). Dotted line indicates lower limit of assay detection.*

Accordingly, the title of the manuscript (Delayed induction of type I and III interferons *mediates* nasal epithelial cell permissiveness to SARS-CoV-2) and the text has been adjusted to modified to reflect these additional data, which show that the delayed IFN response, once established, begins to impact SARS-CoV-2 replication (text changes have been summarised in the response to comment 12).

Reviewer 2

In this manuscript, Hatton et al. studied SARS-CoV-2 infection and the subsequent interferon response of nasal epithelium using an organotypic air-liquid interface model. The authors applied single-cell RNA sequencing to determine the infection of different epithelium cell subsets. In addition, they determined the production of type I and III IFNs, and the subsequent ISG response over time. Finally, they showed that recombinant IFN- β or IFN- λ 1 restricts viral replication.

The manuscript is written in a clear and understandable manner. Nasal epithelium is very relevant tissue to study in the context of SARS-CoV-2, since it is probably the main route of initial infection. The used organotypic model is elegant, and closely mimics in vivo differentiated cells.

The findings that SARS-CoV-2 can infect nasal epithelium, and that this can be partially counteracted by prior treatment with recombinant type I and III IFNs, are very much in line with previous findings and general expectations. The main novelty of this manuscript (as also stated in the title) would be the finding of a delayed type I and III IFN response by nasal epithelium upon SARS-CoV-2 infection.

We thank the reviewer for these supportive comments regarding the experimental model and potential novelty of the observations.

Yet, this reviewer has doubts whether the data support this conclusion of the authors, for the following reasons:

1. First, for many viruses, substantial levels of infection are required before intracellular microbial structures rise above the threshold of detection in epithelial cells. As shown in Fig. 1E, it takes 24h to reach substantial levels, which peak at 48h. In that regard, it is questionable whether the IFN response by these cells is really delayed.

We address this comment alongside the related comment 2 (below), with thanks to the reviewer for both comments.

As recommended in point 2, we conducted additional studies comparing SARS-CoV-2 with other respiratory RNA viruses, influenza A (IAV) and parainfluenza virus 3 (PIV3), at MOI 0.1 (new Fig. S7) and MOI 2 (new Fig. 4E-F).

New Fig. S7

Figure S7. Delayed induction of IFNs and ISGs in response to SARS-CoV-2 compared to other viruses. RT-PCR analysis of IFNB, IFNL1, USP18 and RSAD2 expression in nasal ALI cultures mock infected (0h) or exposed to SARS-CoV-2 (open bars), influenza A virus (IAV H1N1, purple bars) or parainfluenza 3 virus (PIV3, orange bars) for the times displayed, all at MOI 0.1 (n=3 donors, mean ± SEM; ANOVA with Dunnett's post-test correction compared to 0h, or Sidak's post-test correction [all viruses compared at 24 hpi], ** P < 0.01 *** P < 0.001 **** P < 0.0001). ND = not detected.

New Fig. 4E-F

Nasal ALI cultures were infected with SARS-CoV-2 or influenza A virus (IAV H1N1, purple bars) at MOI 2. Whole-cell lysates were prepared at the indicated times for RT-PCR analysis of expression of (E) IFNB, IFNL1 and (F) the ISGs USP18, RSAD2 and ISG15 (n=3 donors, mean +/- SEM; * P < 0.05, ** P < 0.01, *** P < 0.001, **** P < 0.0001, ANOVA with Dunnett's post-test correction compared to 0h).

These experiments demonstrated significant induction of IFN-I/III genes at 6 and 24 hpi with PIV3 and IAV, contrasting with the minimal IFN-I/III response to SARS-CoV-2. These data are consistent with our previous findings with SeV (now Fig. S8) and published studies in different epithelial cell systems (see revised manuscript for full list of citations) and are compatible with the notion that induction of innate IFNs is delayed in response to SARS-CoV-2 relative to other respiratory viruses rather than an intrinsic property of nasal epithelial cells.

It has also been suggested that a greater IFN response to SARS-CoV-2 is achieved with higher multiplicity of infection, at least in cell lines (see Blanco Melo and colleagues, Cell 2020, doi:10.1016/j.cell.2020.04.026). This is in keeping with the reviewer's suggestion that a threshold level of microbial genome content needs to be exceeded to provoke IFN expression. Therefore, we also undertook experiments with IAV and SARS-CoV-2 at 20-fold higher MOI (MOI 20), extending the analysis to 24 hpi. However, the previously observed relative defect of IFN response to SARS-CoV-2 remained apparent (see Fig. 4E-F above).

Finally, in keeping with this delayed kinetic, we observed in additional experiments with the JAK inhibitor ruxolitinib (see comment 16 from Reviewer 1 above) that blockade of IFN signalling

did not begin impacting SARS-CoV-2 replication until 96 hpi (revised Figure 6B-E). This timepoint is approximately 24 h after ISG protein products are robustly detected by immunoblot and mass spectrometry analysis, but at least 48h after viral infection has peaked.

Collectively, these data support the interpretation that the IFN-I/III response to SARS-CoV-2 is delayed relative to viral replication.

2. Second, and more important, it is completely unclear whether this (supposed) delayed IFN response is specific for SARS-CoV-2. The response to SARS-CoV-2 was only compared to one other virus, Sendai virus, which is not only very different from coronaviruses, but also very different regarding pathogenicity. Before the authors can conclude that SARS-CoV-2 shows a delayed IFN response by nasal epithelium, it is essential that they compare this response to IFN production by other (related) viruses.

Again we are grateful to the reviewer for this suggestion, discussed in our response to comment 1 above.

3. In addition, it would be nice if the authors could try to put the findings of this study into perspective, particularly as to whether it would be related to disease severity. Patients only progress to severe disease at about two weeks post infection. Would that still be related to differences in infection and/or IFN responses in nasal epithelium? Are differences in infection and/or IFN responses to be expected between individuals, e.g. pediatric vs adult, related to obesity, or other risk factors?

We appreciate this suggestion and have included a paragraph in the revised discussion reflecting on the potential relationship between IFN responses in the nasal epithelium and disease severity and how this can be modified by risk factors such as age, smoking, etc:

“IFN-I/III signalling is plausibly implicated in protection against life-threatening COVID-19¹⁸⁻²⁰. Consistent with this, circulating immune cells of patients with severe COVID-19 exhibit impaired ISG responses⁵⁵⁻⁵⁷. However, whether the local airway IFN response in the early stages of infection has a decisive role in shaping the subsequent clinical outcome of COVID-19 remains to be conclusively determined. A compelling recent scRNA-seq study reported that patients going on to develop severe disease exhibited a muted ISG response in the nasal airway, in contrast to those with milder disease¹⁵, and is supported by independent findings of attenuated nasal ISG induction in patients with autoantibodies to IFN-I⁵⁸, who are prone to more severe disease²⁰. Additional strands of evidence suggest a potential link between airway IFN-I/III competence and clinical outcome in COVID-19. Age remains the strongest risk factor for poor outcome in COVID-19, and the efficiency of IFN-I/III induction is known to decline with advancing age⁵⁹, and appears to be greater in the nasal airways of children than adults infected with SARS-CoV-2⁶⁰. Other relevant environmental influences, such as exposure to cigarette smoke or other viral infections, are also reported to perturb IFN-I/III responses of airway cells in ways that may be relevant to COVID-19 pathogenesis^{43,61}.”

Reviewer 3

In this work, Hatton and co-workers (NCOMMS-21-06326) employ scRNAseq on SARS-CoV-2 infected ALI cultures to address the antiviral response which is generated by primary airway epithelial cells. They show that most cell types are susceptible to SARS-CoV-2 infection and that

at 24hpi, infected cells produce ISGs while the neighbours non-infected cells (bystander cells) do not. The authors suggest that this limitation of ISG production to infected cells might be the consequence of both a limited production of IFN which appear limited to a small fraction of ciliated cells and a delayed production of IFN (scRNAseq performed at 24hpi). They suggest that the ISG produce in the infected cells are direct ISG, made directly downstream IRF3 without the need of IFN receptor-mediated signalling. Using mass spectrometry approaches, the authors could show that SARS-CoV-2 infected ALI cultures mount a “typical” IFN mediated response (production of classical ISG) at 72 hpi. Finally, the authors show that treatment of cells with RUX to block the IFN-mediated response did not render cells more permissive to SARS-CoV-2 infection while pre-treatment of ALI cultures with IFN partially protected against infection. Together the authors suggest that the delayed IFN response of ALI cultures to SARS-CoV-2 infection is not sufficient to protect the cells against SARS-CoV-2.

While I appreciate the novelty in using scRNAseq on infected ALI cultures, I fail to identify what are the new findings in this work. The fact that most cell types could be infected by SARS-CoV-2 has been previously described and the authors refer to some of these papers and similarly, the fact that the response to viral infection leads to a delay production of IFN which fail to control SARS-CoV-2 was already reported in primary airway epithelial cells culture (Rebendenne et al. 2021). The authors observe that at 24hpi, infected cells produce ISGs while they failed to detect IFN (or very little). The authors speculate that these ISGs are direct ISG made directly downstream IRF3. This is potentially really interesting, but the authors should demonstrate this hypothesis and identify how can at early time post infection, ISG are made directly downstream IRF3 while at later time post infection (48hpi Figure 6), ISG will be made downstream the IFN receptors.

We thank the reviewer for this summary, emphasising the key observation that induction of a paracrine IFN-I/III response to SARS-CoV-2 is robust but kinetically delayed in nasal epithelial cells.

Major comments:

1. Using single cell RNAseq, the authors show that they were able to recover the major cell types of upper respiratory tract (Ciliated, Secretory, Basal etc) based on their gene expression profiles and immunostaining of selected marker genes. Line 108: what do the authors mean by corresponded broadly to ex-vivo scRNA seq data. It is important that the authors perform a qualitative and quantitative evaluation of the gene expression profiles of their ALI culture compared to biopsies, to judge how similar their ALI models are compared to primary airway epithelial cells in-vivo. A systematic comparison with cell types identified from previously published single cell RNAseq studies of primary upper respiratory tract tissues is missing. Such an analysis can be easily implemented with label transfer methods like those implemented in Seurat to exhaustively annotate cell types based on prior knowledge from related primary tissues

We thank the reviewer for this suggestion. We have used an external dataset of scRNAseq of ex vivo nasal swabs (Ziegler et al, Cell 2020, doi:10.1016/j.cell.2020.04.035) and, as the reviewer suggested, performed analysis with the Seurat label transfer tool to compare our annotations.

The results of this analysis, which shows robust correlation with *ex vivo* cell types, are presented in a new Fig. 1B.

- In Line 150 - the authors write - "A greater proportion of ciliated and secretory cells expressed viral transcripts compared to basal cells", however looking at Fig 2(A,B), the cell subtype FOXN4 also seems to express the viral transcripts at similar levels, in-fact higher than secretory cells and at similar levels to ciliated. Surprisingly, this rare cell type is hardly discussed in the manuscript. The authors should explain the reasons behind this. A complementary study from Ziegler et. al, <https://doi.org/10.1101/2021.02.20.431155> (Bioarxiv preprint), has shown that the FOXN4+ cells are affected upon SARS-Cov2 infection, thus in my opinion, this cell type warrants further investigation in the ALI infection model system

We are grateful to the reviewer for pointing out this oversight. As described in the responses to reviewer 1, data were re-analysed following CellBender processing. Based on differential gene expression analysis between infected cell types (Wilcoxon rank sum test, one vs rest, supplementary dataset 2), secretory and ciliated cells expressed higher levels of viral transcripts compared to other cell types, with viral transcripts most abundant in secretory cells. Following removal of ambient RNA signal with CellBender, the expression of SARS-CoV-2 transcripts in the FOXN4+ (deuterosomal) population is no longer significantly greater than in other cell types. We include these results of this DE analysis as a supplementary dataset 2 and include the revised text (results):

"While all cell types expressed viral transcripts, there were notable differences both in the proportion of cells infected, and the relative abundance of different viral transcripts within these cells (Fig. 2A). Based on differential gene expression analysis between infected cell types (Wilcoxon rank sum test, one vs rest, $P < 0.05$), secretory and ciliated cells expressed higher levels of viral transcripts compared to other cell types, with viral transcripts most abundant in secretory cells (Fig. 2A, supplementary dataset 2)."

We also highlight the infection of deuterosomal cells in the revised discussion:

“Nevertheless, our data, employing a variety of complementary methods, indicate that SARS-CoV-2 has a relatively broad tropism for nasal epithelial cells, confirming suggestions from prior scRNA-seq studies^{8,10}, other in vitro studies of primary nasal¹⁴ and tracheobronchial cells^{24,35}, and importantly recent scRNA-seq studies of nasal samples from COVID-19 patients¹⁵. We also identify tropism for the rare deuterosomal cell, marked by expression of FOYN4, as recently reported^{35,58,63}.”

See also the response to reviewer 1, point 11. In view of their rarity, our ability to assess responses further in this cell type was relatively limited.

3. INF gene sets - The authors have used a generic ISG gene list from MsigDB to score the interferon response in unexposed, uninfected and infected cells, however, interferon-stimulated gene lists specifically from primary nasal epithelium tissues and ALI models treated with different interferons in dose dependent manner exists. Using such context specific ISGs is probably better to evaluate the immune response generated by the ALI culture.

We are very grateful for this helpful suggestion. As recommended, we used published context specific IFN α and IFN γ gene lists (primary nasal cells treated with cytokines, doi: doi:10.1016/j.cell.2020.04.035] to annotate ISGs in revised Fig. 3A and 3B.

A

B

C

Pathways regulated in infected cells

An interesting observation emerged from this analysis, in that basal cells appear to mount a more efficient IFN-I/III response than other cells, as described in the results:

“Basal cells expressed a modest number of ISGs upon infection, specifically genes of the IFITM family, IFI27 and IFI6 and the negative regulator of IFN-I signalling, ISG15. Consistent with this finding, gene-set enrichment analysis (GSEA) revealed upregulation of IFN alpha response in infected basal cell populations but not in other cell types (Fig. 3C).”

This observation is discussed in the revised manuscript (discussion):

“The differential response of basal cell types to SARS-CoV-2 at 24 hpi identified by our scRNA-seq analysis appears a novel observation. Basal cells are the stem/progenitor cell population of the airway⁶⁴. Recent data indicate an emerging role for these cells as sentinels of the airway inflammatory response⁶⁵. For example, basal cells detect apoptotic cells in the context of viral inflammation⁶⁶, retaining memory of prior immune exposure⁶⁷. More generally, stem/progenitor cell types exhibit enhanced intrinsic antiviral immunity⁶⁸. Future studies should consider mechanism(s) governing the seemingly distinct early antiviral response of nasal airway basal cells to SARS-CoV-2, and its functional relevance.”

4. Instead of simply looking at the expression of regulators of INF response, it might be insightful to perform gene regulatory analysis using methods like SCENIC to identify the transcription factor activity of relevant TF's like IRF's, JAK/STATS in the infected cells and corroborate some of those with the proteomics results.

As recommended, we have performed gene regulatory analysis using the DoRoThea package and include these data in revised Fig S5.

Figure S5. DoRoThea/VIPER analysis of regulon activity in infected cells. Median regulon activity per cluster in infected cells, corrected for activity in uninfected cells by subtraction then Z-normalised by TF

(i.e. values > 0 imply TF more active in infected cells). Data from analysis of 28,346 cells total to estimate regulon activity of which 8,861 infected, from n=2 donors at 24 hpi.

This analysis predicts NF- κ B2, STAT4 and STAT6 TFs are more active in all cell types upon infection. Comparing TF activity between cell types, activity of most JAK/STAT TFs is highest in secretory cells. As a complement to this analysis, we also conducted unbiased GSEA and investigated the results via pathway analysis. This indicated differential regulation of a range of host response pathways, including NF- κ B signalling, as summarised in revised Fig. 3C (see above in response to point 3), but did not identify enrichment of IFN response pathways in secretory cells, consistent with the DE analysis.

The probable explanation for the apparent lack of correlation with the proteomics data (at 72 hpi) pointed out by the reviewer is the delayed kinetic of the IFN response, as demonstrated in Fig. 4A-D and Fig. 6A. See also the response to comment 6 below.

5. It is very difficult to appreciate the immune response generated by the different cell types from the heatmap shown in figure 3B. I am expecting the data to be the relative expression compared to mock infected cells. I could not find information about the expression value, these are Log values? Figure 3B should be complemented with a supplementary figure that plot all ISG expression level for a cell type (with the X-axis being mock, bystander and infected). This will allow the reader to have a better view of ISG expression and whether they are up or downregulated upon infection. From the scRNAseq data, it looks like the level of some ISG expression is even down regulated in the bystander cells. The authors should explain this rather unusual inhibition of ISG production in bystander cells? Can SARS-CoV-2 inhibit IFN-mediated response in bystander cells by secreting viral/cellular factors? This should be address experimentally.

We apologise for this lack of clarity in the presentation of Fig. 3B. Accordingly we have revised the presentation of this figure, utilising volcano plots, and annotating context-specific ISGs. Upon reanalysis using CellBender during data processing (see response to comment 3, in addition to reviewer 1, major comment 1) and annotation with context-specific ISGs there is no evidence of widespread ISG downregulation in bystander cells, with the exception of *IFITM* genes (Fig. 3B, see below). The significance of this isolated reduction in *IFITM* genes is presently unclear, although we note that IFITMs have been implicated in entry of SARS-CoV-2. This is mentioned in the revised text:

“Interestingly, IFITM genes (ISGs which have been paradoxically implicated in SARS-CoV-2 entry³³) were downregulated in some bystander cell populations.”

Fig. 3B

- the authors suggest that the fact that only infected cells display ISG at 24hpi in their scRNAseq data is the consequence of direct ISG production (downstream IRF3). According to the data shown in figure 6 using higher MOI and different time post-infection, the lack of ISG in bystander cells may be due be the result of the time point used for the scRNAseq (24hpi), too soon to detect paracrine signalling. The authors should rule out this possibility. For example, the experiment shown in figure 6 with RUX should be repeated at earlier time post-infection to show that the production of ISG in the infected cells is not due to IFN-mediated signalling but to a direct production of ISG via IRF3.

We thank the reviewer for this suggestion. Please note that we did not originally intend to propose direct ISG production to be the dominant mechanism, nor did we explicitly state this in the original manuscript. Indeed, we saw very minimal induction of ISGs by bulk analysis methods (RT-PCR or immunoblot) at earlier times (i.e. 6 hpi or 24 hpi) and this is born out in additional studies done at higher MOI (see response to reviewer 2 above). Consistent with this, upon re-analysis with CellBender during data processing there was no evidence of uniform ISG

induction by nasal cells at 24 hpi (see Fig. 3B panel displayed above and discussion in response to point 5).

We agree with the reviewer's suggestion that the timepoint is likely an important factor in the ability to detect ISGs in bystander cells. As recommended by the reviewer, we accordingly repeated the experiment originally shown in Figure 6 using RUX at earlier as well as later times post infection, using RT-PCR to detect ISG expression (see new Fig. 6A).

This experiment demonstrated that induction of the ISGs *ISG15*, *USP18* and at least at earlier times *RSAD2* was IFN-dependent, and therefore did not support a role for IRF3 in ISG induction at 24 hpi. Consistent with this, we observed no induction of ISG products at the protein level in RUX treated cells up to 96 hpi (Fig. 6C).

C

7. Why do the authors only focus on type I IFN for the exogenous IFN treatment experiment at different time pre- and post-infection? The authors should also add exogenously type III IFN to address which IFN (Type I or type III OFN) would be best as therapeutic approaches

Again, we are grateful for this suggestion. As requested, we have included additional experimental data on the antiviral activity of IFN-III in a new Figure 7 (panels D and E).

Our findings now show that exogenous Type I or III IFNs can be used to suppress SARS-CoV2 infection of nasal epithelial cells in early times after infection. These data suggest that both IFN-I and IFN-III would be suitable therapeutically.

Minor comments:

- Line 125: the decline of TEER can also be due to cell death, the authors should add a little statement about this in their text. Could the authors provide the non-cut version of their WB in supplementary? This will be very useful for reference purposes for other lab to appreciate the background of the antibodies used the authors should label their plots with the name of the different cell types but not the cell type specific markers as it will make it easier for the reader to quickly know what cell type the plot is about.

We agree and include a comment in the text about the possibility of cytopathicity as an explanation of the loss of transepithelial resistance as suggested.

"SARS-CoV-2 replication was accompanied by a progressive decline in epithelial barrier integrity starting from 48 hours post-infection (hpi), reflecting virus-induced epithelial dysfunction and/or potential cytopathic effect (Fig. 1H)."

We also include uncut versions of all blots in the source data file.

Finally, as requested, we have changed the plots so the cell type, rather than marker, is identified (Fig. 2C-D, Fig. S4). However we leave the marker description in Fig. S2 in the interests of informing the reader.

2. line 302, on the WB figure 6A, there is a greater band for the lower one in the presence of RUX. Is this non-glycosylated spike? Because the amount increase with RUX.

The antibody (raised against a C-terminal immunogen) consistently recognises a lower MW band, corresponding to the proteolytically cleaved S2 subunit of the spike glycoprotein (see Nunes-Santos et al Journal of Clinical Immunology, 2020). Accordingly, we have revised the relevant figures (Fig 1F, 4D, 6C) to display both spike and S2.

We have also undertaken densitometry analysis to enumerate changes in the total abundance of spike/S2 protein in revised Figure 6D at 96 hpi (see below). This analysis (Fig. 6C-D) showed a significant increase in S/S2 expression, consistent with a significant increase in viral RNA expression (Fig. 6B) and release of infectious virus (Fig. 6E) at this timepoint, indicative of the ability of IFN-I/III signalling to impact on viral replication at later timepoints. To our knowledge, this has not been previously been shown in nasal cells.

3. Data availability - Given that the data generated in this study will be very useful to the larger community, in addition to submitting the raw data to standard repositories, the authors should also consider making their processed data from their single cell RNAseq analysis (counts and cell annotation tables as Seurat objects) and Proteomics analysis (Normalized log₂ transformed tables as CSV table) easily available. These non-restricted data can be provided as supplementary material or through third party hosting options like Figshare or Github.

The proteomics data were provided in the original submission as supplementary dataset S1.

Single cell data analysis and code is available at Github:

https://github.com/haniffalab/covid_nasal_epithelium

We also include the processed RNA-seq data at Zenodo (<https://zenodo.org/record/4564332>) and raw data are submitted to the PRIDE and EGPA repositories respectively.

These changes are summarised in the revised data and code availability statements:

Data availability

Source data are provided with this paper. This includes uncropped blots, all quantitative data and the results of differential expression analysis of RNA-seq and proteomics data, which are included as supplementary datasets. Additional raw data are available on request from the corresponding author providing ethical approvals permit sharing of data. The mass spectrometry proteomics data have been deposited to the ProteomeXchange Consortium⁸⁰ via the PRIDE partner repository⁸¹ with the dataset identifier [PXD022523](https://doi.org/10.6019/PXD022523). Raw RNA sequencing data have been deposited to the European Genome-phenome archive (accession pending). Processed scRNAseq data is available at Zenodo (<https://zenodo.org/record/4564332>).

Code availability

Analysis scripts and codes are available at github.com/haniffalab/covid_nasal_epithelium.

4. Data Reproducibility - In interests of data reproducibility, transparency and open science, the authors should also share their scripts for performing the computational analysis related to single cell RNAseq and proteomics experiments shown in this manuscript in a code sharing platform like GitHub/GitLab etc

See comment above.

5. Standard QC metrics from Cell ranger (number of reads, reads per cell, alignment rate, cell counts etc) and QC plots from Seurat (nCount, nUMI, %MT etc) pertaining to the single cell RNAseq samples should also be provided as supplementary tables.

These data are now included in a new Fig. S1 and supplementary Table S1.

Figure S1. Single cell RNA sequencing quality control plots. Violin plots, split by sample, showing (A) the total number of genes detected in each cell (B) the total number of counts detected in each cell and (C) the proportion (as a percentage) of mitochondrial transcripts in each cell. For individual QC metrics see also Table S1.

Sample_id	Total number of	Mean reads per	Alignment rate (%)	Reads mapped to	Reads mapped to	Estimated number of
-----------	-----------------	----------------	--------------------	-----------------	-----------------	---------------------

	reads	cell		GRCh38 (5)	SARS-CoV-2	cells
Donor4_Mock	184,897,338	14,329	91.3	91.3	0	12,904
Donor4_Infected	807,865,486	61,100	87.7	77.1	10.8	13,222
Donor6_Mock	232,499,890	17,121	90.4	90.4	0	13,580
Donor6_Infected	174,974,839	13,653	90	88.8	1.3	12,816

Table S1. Single cell RNA-seq post-alignment quality control metrics. Quality control output from CellRanger following alignment.

REVIEWERS' COMMENTS

Reviewer #1 (Remarks to the Author):

This review was prepared and is signed by Dr. Jose Ordoñas-Montanes.

In the revised manuscript by Hatton et al., the authors have further strengthened their conclusions related to the identification of the SARS-CoV-2 viral targets in human nasal ALI cultures, and the role of the kinetics of the IFN response in controlling infection. They have improved the computational methods utilized to analyze scRNA-seq data, adding further clarity to their results, especially through the use background correction methods. This re-analysis highlights secretory cells appearing as the largest fraction of infected cells while ciliated cells contain the greatest abundance of virus. This agrees with recent in vivo data from scRNA-seq studies. The additional quantification and inclusions of controls for microscopy data has helped. The addition of IAV and PIV3 challenges at different MOIs strengthens the delayed kinetics of SARS-CoV-2 using human-relevant viruses, and the addition of later timepoints in the ruxolitinib experiments strengthens the conclusions related to delayed timing and the impact of the IFN response on controlling SARS-CoV-2 replication. The discussion has been substantially improved with additional context. The major conclusions raised by the authors in the abstract and throughout the paper are well substantiated by their data and the code and data have been shared. There are only a few minor comments remaining for the authors to address:

1. Minor comments

- a. Dot plot in Fig 1c suggest relabel Foxn4 to deuterosomal for consistency.
- b. Line 315-317: word choice for basal expression suggest baseline expression to not confuse with basal referring to basal cells.
- c. Is the western blot legend above Figure 6c correct or is the RUX+ condition shifted over?
- d. Line 977 NovoSeq to Illumina NovaSeq.

Reviewer #2 (Remarks to the Author):

I thank the authors for addressing my questions so thoroughly. They now indeed provide very clear evidence for delayed IFN responses to SARS-CoV-2 by nasal epithelium.

This work nicely fits the bigger picture of reduced IFN responses and severe disease. I am very much looking forward to their future work, in which they may compare the response of nasal epithelium in the context of currently known risk factors.

Reviewer #3 (Remarks to the Author):

see comments to editor

Dear Nature communications editorial team

The authors have properly answered my technical comments, however, I am still not very excited about this manuscript. As mentioned in my previous communication, I do not see what this paper is bringing to an already saturated field.

Most importantly, in response to one of my comment, they performed additional experiments to challenge their model that the IFN response was not able to efficiently protect the cells from infection. By testing different times infection upon inhibition of the immune response, they now can see that blocking the immune response favors viral infection (which was expected).

These novel findings, completely change their model and I invite you to look at their abstract in the first version and in this revised version. the conclusion is now opposite and the immune response via paracrine signaling impact SARS-CoV-2 infection.

best regards

RESPONSE TO REVIEWERS' COMMENTS

Reviewer #1 (Remarks to the Author):

This review was prepared and is signed by Dr. Jose Ordovas-Montanes.

In the revised manuscript by Hatton et al., the authors have further strengthened their conclusions related to the identification of the SARS-CoV-2 viral targets in human nasal ALI cultures, and the role of the kinetics of the IFN response in controlling infection. They have improved the computational methods utilized to analyze scRNA-seq data, adding further clarity to their results, especially through the use background correction methods. This re-analysis highlights secretory cells appearing as the largest fraction of infected cells while ciliated cells contain the greatest abundance of virus. This agrees with recent in vivo data from scRNA-seq studies. The additional quantification and inclusions of controls for microscopy data has helped. The addition of IAV and PIV3 challenges at different MOIs strengthens the delayed kinetics of SARS-CoV-2 using human-relevant viruses, and the addition of later timepoints in the ruxolitinib experiments strengthens the conclusions related to delayed timing and the impact of the IFN response on controlling SARS-CoV-2 replication. The discussion has been substantially improved with additional context. The major conclusions raised by the authors in the abstract and throughout the paper are well substantiated by their data and the code and data have been shared.

We thank the reviewer for this supportive comment.

There are only a few minor comments remaining for the authors to address:

1. Minor comments

a. Dot plot in Fig 1c suggest relabel Foxn4 to deuterosomal for consistency.

This has been corrected.

b. Line 315-317: word choice for basal expression suggest baseline expression to not confuse with basal referring to basal cells.

Changed to baseline.

c. Is the western blot legend above Figure 6c correct or is the RUX+ condition shifted over?

This has been corrected.

d. Line 977 NovoSeq to Illumina NovaSeq.

This has been corrected.

Reviewer #2 (Remarks to the Author):

I thank the authors for addressing my questions so thoroughly. They now indeed provide very clear evidence for delayed IFN responses to SARS-CoV-2 by nasal epithelium.

This work nicely fits the bigger picture of reduced IFN responses and severe disease. I am very much looking forward to their future work, in which they may compare the response of nasal epithelium in the context of currently known risk factors.

We thank the reviewer for these comments.

Reviewer #3 (Remarks to the Author):

see comments to editor

Dear Nature communications editorial team

The authors have properly answered my technical comments, however, I am still not very excited about this manuscript. As mentioned in my previous communication, I do not see what this paper is bringing to an already saturated field.

Most importantly, in response to one of my comment, they performed additional experiments to challenge their model that the IFN response was not able to efficiently protect the cells from infection. By testing different times infection upon inhibition of the immune response, they now can see that blocking the immune response favors viral infection (which was expected).

These novel findings, completely change their model and I invite you to look at their abstract in the first version

and in this revised version, the conclusion is now opposite and the immune response via paracrine signaling impact SARS-CoV-2 infection.

We thank the reviewer for this assessment. As the reviewer points out the data show that JAK inhibition impacts viral infection at very late times and the abstract was amended accordingly.